

# Review article: Analysis of sediment disaster risk assessment surveys in Brazil: A critical review and recommendations

Thiago D. dos Santos[1,2], Taro Uchida[3]

[1]Geological Survey of Brazil, Rio de Janeiro 22290-255, Brazil

[2]Doctoral Program in Environmental Studies, Institute of Life and Earth Sciences, Graduate School of Science and Technology, University of Tsukuba, Tsukuba 305-8577, Japan

[3]Institute of Life and Environmental Sciences, Graduate School of Science and Technology, University of Tsukuba, Tsukuba 305-8577, Japan

*Correspondence to*: Thiago D. dos Santos (tdutra.18@gmail.com)

**Abstract.** Climate change-induced sediment-related disasters in Brazil are intensifying, posing substantial risks. Studies on Brazilian disaster risk reduction are abundant, but those on federal risk assessment surveys are scarce. To address this gap, we analyzed five surveys, including the Municipal Risk Reduction Plan (PMRR), Geological Risk Survey (GRS), Susceptibility Survey (SS), Geotechnical Aptitude for Urbanization (GAUC), and Geological Hazard Survey (GHS). We conducted a meta-analysis of 300 scholarly publications and public datasets to assess these surveys, evaluating input data, methods, outcomes, 15 applicability, effectiveness, and cost–benefit, guided by global recommendations. Spearman's rank correlation and McDonald's Omega were employed to evaluate survey associations with initiatives. The results reveal each survey's unique contributions and challenges, such as limited national coverage and underutilization of quantitative methods. GHS stands out for its versatility, including climate change adaptation countermeasures and decision-maker relevance, but it lacks legal support and limited coverage. GRS and SS are well established but need considerable methodological updates, while GAUC is 20 underutilized due to complexity and high costs. Despite the reproducibility and cost-time efficiency challenges, PMRR exhibits substantial correlation with implementing disaster risk reduction activities. Recommendations include standardizing procedures, enhancing data collection and analysis, improving outputs, and a progressive multilevel approach.

## 1 Introduction

The United Nations Office for Disaster Risk Reduction (UNDRR) defines a disaster as a significant disruption in a 25 community's normal functioning, marked by extensive losses in human lives, property, the economy, and the environment. It arises from a combination of hazard exposure, vulnerability, and inadequate measures to mitigate potential negative impacts (UNDRR, 2017). Disaster risk reduction (DRR) focuses on reducing the impact of socioeconomic disasters through systematic assessment, mitigation, preparedness, response, and recovery efforts. Fell et al. (2008) suggested that regional, local, and site-specific risk assessment surveys are vital for establishing an effective disaster preparedness and response framework. This





framework must be tailored to align with local specificities and needs, addressing practical challenges and enhancing community resilience against sediment disasters.

As a continental and diverse nation encompassing various geographical and climatic regions, Brazil is inherently susceptible to numerous landscape-altering natural phenomena that lead to catastrophic events (Pimentel et al., 2020). From colonial times to the present day, the country has faced numerous natural events that have affected millions of people's lives and economies

(e.g., De Ploey and Cruz, 1979; Schuster et al., 2002; Mello et al., 2014). The IPCC (2023) report indicates that ongoing climate change intensifies the frequency and severity of extreme weather events, markedly affecting human health, livelihoods, and critical infrastructures, especially in urban environments.

According to Giustina (2019), Brazilian DRR Management, as outlined by the National Policy for Civil Protection and Defense (Brasil, 2012), includes an initial assessment phase to identify hazardous sediment disaster-prone areas and at-risk communities

by analyzing previous events, soil, topography, hydrology, and socioeconomic conditions; this phase supports subsequent steps and activities. Preparedness entails educating communities and establishing early warning systems (EWSs) and emergency response plans (ERPs). The emergency phase focuses on immediate assistance, including evacuations, search and rescue operations, and temporary shelters. Post-disaster recovery aims to reestablish safety, rebuild infrastructure, reinforce mitigation measures, and support affected communities. The mitigation phase involves implementing strategies to reduce the likelihood

and severity of such disasters through engineering solutions and land-use policies.

The municipal master plan (MP), established by Brazil's Law No. 10,257 (Brasil, 2001) and modified by Law No. 12,608 (Brasil, 2012), is an essential document for urban planning, DRR initiatives, and long-term sustainable development strategies (MDR, 2021). MP is compulsory and required in municipalities exceeding 20,000 inhabitants.

The latest Brazilian Atlas of Natural Disasters (Brasil, 2023) reports a significant rise in geohydrological disasters from 1991

to 2023, primarily floods and sediment-related events, accounting for 82% of the death toll. The 21,043 disasters have claimed 3,752 lives, left over 7.45 million homeless, and impacted nearly 77 million, either directly or indirectly. The economic cost is estimated to be approximately USD 19.63 billion.

Over the decades, significant global progress has been made in sediment-related risk management, including phenomena typology and classification (e.g., Varnes, 1978; Cruden and Varnes, 1996; Hungr et al., 2001; Hungr et al., 2014; Li and Mo,

2019). Insights into the predisposing, triggering, and dynamic factors influencing these events abound (e.g., Guzzetti et al., 1999; Iverson, 2000; Dai et al., 2002; Hungr, 2007; McColl, 2022; Iverson, 2015; McDougall, 2017). Additionally, considerable advancements have been achieved in assessing these phenomena (e.g., IAEG, 1990; Corominas, 1996; Aleotti and Chowdhury, 1999; Guzzetti et al., 1999; Corominas et al., 2003; Picarelli et al., 2005; Corominas and Moya, 2008; Fell et al., 2005; 2008; Van Westen et al., 2008; Corominas et al., 2014; Davies, 2015; Hungr et al., 2016). Moreover, in recent years,

innovative approaches have been developed to enhance preparedness (Colombo et al., 2005; Ayalew and Yamagishi, 2005; Uchida et al., 2011; Guzzetti et al., 2012; Chen et al., 2014; Sun et al., 2020; Di Napoli et al., 2021; Linardos et al., 2022). Despite the advancements, Maes et al. (2017) highlighted that in tropical nations, only 30% of potential risk reduction measures are recommended or implemented, with risk assessments emerging as the most frequently implemented initiatives (57%).





Numerous studies have explored the aspects of Brazil's governmental DRR framework. Tominaga et al. (2012) analyzed
Brazil's socioeconomic disasters and risk-management strategies. Ganem (2012), Almeida (2015), and Henrique and Batista
(2020) evaluated DRR's political dimensions. Kuhn et al. (2022) discussed the evolution and impacts of DRR policies, while
Alvalá et al. (2019) studied the profiles of at-risk and vulnerable populations. Silva-Rosa et al. (2015) and Matsuo et al. (2019)
emphasized environmental education in disaster reduction. Mendonça and Gullo (2020) focused on societal disaster perception,
Marchezini et al. (2019) on education's role in risk mitigation, and Silva and Santos (2022) on the importance of social
participation in DRR. Debortoli et al. (2017) and Marengo et al. (2021) discussed the impact of climate change on Brazilian
disasters, highlighting the need for climate-inclusive DRR strategies. Silva (2022) extensively analyzed federal DRR projects
from 2011 to 2015, offering a broad view of government initiatives in this area. Specifically, for risk assessment, Mendonça
et al. (2023) evaluated the effectiveness of Municipal Risk Reduction Plans (PMRR) in Brazil. Dias et al. (2021) examined
landslide susceptibility mapping methods, while Rocha et al. (2021) compared the accuracy of three national survey outcomes
(Susceptibility - SS, Geotechnical Aptitude Urbanization - GAUC, and Geological Hazard – GHS surveys) in Nova Friburgo
city, Rio de Janeiro. However, no previous studies have exclusively focused on the core elements of federal risk assessment
surveys nationwide. Hence, this study aims to bridge this gap by analyzing the nuances of these surveys, identifying
methodological deficiencies, and proposing improvements, thus enhancing Brazilian strategies for a more resilient society.

## 2 Materials and Methods

In this study, we conducted a meta-analysis using 300 scholarly publications selected from Google Scholar based on their
citations and relevance. From 1979 to 2024, these publications provided information related to Brazil's sediment-related
disaster history and DRR initiatives, such as legislative frameworks, governmental directives, surveying data, and existing
articles. In addition, we supplemented this with government reports and newsletters. Moreover, we systematically analyzed
publications on globally acknowledged best practices to assess the surveys and critically derive well-informed
recommendations.

### 2.1 Data collection and analysis

Various public-domain datasets were employed to analyze the disaster landscape in Brazil. The Brazilian Atlas of Natural
Disasters, from 1991 to 2023, served as a primary source of official disaster reports (Brasil, 2023). The analysis was further
narrowed down to incidents related to sediment disasters. The municipalities highly susceptible to sediment-related disasters
were retrieved from the Ministry of Regional Development—National Secretariat of Civil Protection and Defense (MDR,
2012), pinpointing areas requiring particular focus for conducting DRR assessment surveys. The 2022 demographic dataset
and current geopolitical and socioeconomic metrics were obtained from the Brazilian Institute of Geography and Statistics
demographic census (IBGE, 2022). The DRR initiative dataset was derived from the Basic Municipal Information Survey
(IBGE, 2020; 2021), covering human resources, legislative frameworks, territorial management, socioeconomic indicators,



and specific risk and disaster management information. The DRR data detailed initiatives for planning, monitoring, response, and municipal infrastructure to address droughts, flooding, erosion, landslides, and slope failures. The datasets were reorganized, categorized, and stratified by major geographic regions and federal units and divided by state and municipal population sizes to obtain a nuanced and region-specific perspective.

## 2.2 Studied risk assessments

A comparative evaluation of the five federal risk assessment methodologies initiated after 2004, including the PMRR, Geological Risk Survey (GRS), Susceptibility Survey (SS), Geotechnical Aptitude for Urbanization Charts (GAUC), and Geological Hazard Survey (GHS), was conducted. Information was collected and reviewed from official guidelines and their updates (Alheiros, 2006; Brasil, 2007; Bittar, 2014; Pimentel and Dutra, 2018; Lana et al., 2021; Antonelli et al., 2020; Antonelli et al., 2021) alongside outcomes (Mendonça et al., 2023; SGB, 2023a; b; c; d), extracting the essential

methodological concepts and procedures. These surveys were analyzed using the assessment method along with the input and output data.

## 2.3 Evaluation methods for the impact of risk assessment surveys

We investigated the impacts of these surveys using a four-component evaluation framework, namely, survey implementation status, applicability to DRR initiatives, progress of these initiatives, and cost-per-beneficiary analysis. This approach assesses

the practical application of these surveys, as outlined by Hungr et al. (2005), their effectiveness in Brazilian states and municipalities using IBGE (2020) data, and their costs in a medium-sized municipality. By integrating these aspects, we aimed to understand each survey's utility, performance, and socioeconomic impact on enhancing DRR strategies.

### 2.3.1 Implementation status of risk assessment surveys

The implementation status of these surveys was analyzed to assess their extent and advancement across the country. Data for

the PMRR were retrieved from the Basic Municipal Information Survey focused on risk management (IBGE, 2020), but the GRS, SS, GAUC, and GHS surveys were obtained from the Geological Survey of Brazil's official repository, the institute responsible for implementing these surveys (respectively, SGB, 2023a; b; c; d). To effectively assess the implementation status, we calculated the national percentage coverage of each survey relative to the total number of municipalities in a state. Geographic information system (GIS) tools were applied to identify spatial distributions and patterns.

### 2.3.2 Applicability for disaster reduction countermeasures

#### – Scales

We adopted the ordinal classification system, proposed by Fell et al. (2008) and Corominas et al. (2014), for scoring, emphasizing customization of assessment scale and scope to match local and state authorities' requirements, data availability,



and survey objectives. Consequently, we categorized the surveys into four distinct assessment scales: National Scale (< 1:
250,000), Regional Scale (1: 250,000 to 1: 25,000), Local Scale (1: 25,000 to 1: 5,000), and Local Scale (1: 25,000 to 1: 5,000).
These scales enable a tailored approach, allowing specific interventions at each level and ensuring practical and effective
outcomes for diverse contexts. Additionally, we adopted an applicability gradient, ranging from dark blue (0—Not applicable)
to dark red (4—Fully applicable), to categorize the survey scales.

**– Parameters**

To assess the applicability of the risk assessment surveys, we consulted various sources, including critical studies by Aleotti
and Chowdhury (1999), Guzzetti et al. (1999), Cascini (2008), Fell et al. (2008), Guzzetti et al. (2012), and Corominas et al.
(2023). These sources provided insight into specific objectives, utilities, and information requirements. We focused on the
most commonly used parameters in strategies across different stages of the DRR cycle. The parameters refer to national and
local legislative frameworks and urban planning for prevention. Preparedness metrics included early warning alert systems,
responsive parameters covered ERPs, and mitigation involved prioritization of structural countermeasures (SC). Given the
increasing impact of climate change (e.g., Marengo et al., 2021), we integrated specific criteria to evaluate the surveys'
contributions to building a more resilient society, as suggested by Bozzolan et al. (2023).

### 2.3.3 Implementation status of disaster reduction initiatives

To assess the effectiveness of risk assessment surveys in enhancing DRR efforts, we analyzed the adoption and implementation
of initiatives across 5,570 municipalities using the IBGE (2020) dataset. We focused on six key initiatives: 1) Prevention,
Master Plan (MP), Municipal Land Use Land Occupation Law (LULOL), and Municipal Landslide Specific Laws (SL); 2)
preparedness through early warning alert system; 3) response via emergency response plan (ERP); and 4) mitigation by
implementing structural countermeasures (SC). These comprise six nonstructural initiatives promoting appropriate land-use
policies (MP, LULOL, and SL), management (EWS, ERP), and one structural measure (SC). The GAUC and GHS were
excluded from this analysis due to their limited national representativeness. Furthermore, we employed two statistical methods
to evaluate the impact of these surveys on DRR initiatives in the states. The analyses were performed using Jamovi, a free and
open-source statistical software for data analysis (Jamovi, 2024).

1)          Spearman's rank correlation coefficient test (ρ) was used for nonparametric data distribution to measure the strength
and direction of the association between the surveys and DRR initiatives (Rezaei et al., 2018). This method ranks the data and
calculates the correlation based on these ranks, making it robust against outliers and suitable for nonlinear relationships using
Eq. (1):

$$\rho = 1 - \frac{6 \sum d_i^2}{n(n^2-1)} \quad (1)$$

where *di* represents the differences in ranks between the two variables for each observation *i*, and n is the total number of
observations. The value of $\rho$ ranges from −1 to +1, where positive values indicate a positive correlation and negative values



indicate a negative correlation. The strength of the correlation is interpreted based on: 0–0.2 (very weak), 0.2–0.4 (weak), 0.4–
       0.6 (moderate), 0.6–0.8 (strong), and 0.8–1 (very strong).

2)    McDonald's Omega (ω) was employed in the reliability testing to quantify the overall commonalities and unique
attributes within the factor structure of survey instruments (McDonald, 1999). This approach is particularly effective for
understanding how items in a survey collectively reflect a single underlying construct. The Omega coefficient is calculated
using Eq. (2):

$$\omega = \frac{(\sum \lambda_i)^2}{(\sum \lambda_i)^2 + \sum \psi_i} \quad (2)$$

where λi represents the factor loadings of the items on the common factor, indicating the degree of correlation between each
item and the underlying construct; ψi denotes the unique variances of the items, capturing the proportion of each item's
variance that is not explained by the common factor. This dual consideration of commonalities and unique aspects enhances
the precision of reliability assessments, ensuring that each item's contribution to the overall construct is accurately measured.
These statistical techniques provide a robust basis for understanding the correlation between surveys and the effectiveness of
implemented DRR initiatives, aiding in the refinement and strategic planning of future strategies.

**2.4 Costs per beneficiary**

According to Heo and Heo (2022), cost-per-beneficiary analysis (CBA) is a vital socioeconomic tool for quantitatively
assessing the efficiency and impact of DRR strategies. This evaluation allows an objective assessment of the financial
investment relative to the direct benefits received by the communities (Price, 2018). A basic CBA approach has been employed
to investigate RA surveys, focusing on medium-sized municipalities in Brazil. The analysis considered the PMRR of São José
dos Campos municipality in São Paulo state (IPPLAN, 2016). The division manager at the Geological Survey of Brazil,
responsible for implementing GRS, SS, GAUC, and GHS surveys, directly provided key performance metrics. All monetary
values originally in Brazilian Reais have been converted into US dollars for enhanced clarity. The CBA ratio was computed
by dividing the average implementation costs by the average number of beneficiaries. A higher cost–benefit ratio means that
the survey is less economically efficient due to a higher cost per beneficiary (Shreve and Kelman, 2014). Conversely, a lower
ratio suggests a better economic efficiency, indicating that the survey benefits more people than the costs incurred. Therefore,
this analysis helps policymakers and stakeholders to identify the most cost-effective surveys, aiding in informed resource
allocation decisions.

**3 Sediment Disaster Risk Assessments Surveys in Brazil**

In Brazil, various classifications for sediment phenomena are used (e.g., Cruden and Varnes, 1996; Augusto Filho, 1992;
Hungr et al., 2014). However, the Brazilian Classification and Coding of Disasters (COBRADE) categorized these phenomena





into four main types: 1) Falling, tilting, and rolling (involving blocks, flakes, boulders, and slabs), 2) landslides (both shallow

and deep-seated), 3) flows (comprising soil/mud and rock/debris), and 4) subsidence and collapses (COBRADE, 2012).

### 3.1 Brief history of sediment disasters and risk assessment

Disasters are a product of political and historical decisions related to territory organization and social coexistence in response

to hazard exposure (Veyret, 2007). To identify and quantify the correlations among key determinants that may convert

sediment-related phenomena into disasters, we analyzed data from the Brazilian Institute of Geography and Statistics (IBGE,

2020; 2022) and assessed the correlation of geopolitical, demographic, and socioeconomic factors, such as the total and urban

areas (km$^2$) of states, population ($10^6$ habitants), demographic density (Hab./Area), and the human development index (HDI)

with disasters and critical municipalities reports within states (Fig. 1). The HDI is a composite statistic that measures a

country's social and economic development across three main dimensions: (1) Life Expectancy, (2) Education, and (3) Income.

Given the nonlinear nature of these relationships, we employed the Spearman coefficient (rho) to conduct a nonparametric

data analysis.

Our analysis revealed significant correlations that aid in understanding disaster risk (Fig. 1). For example, a robust positive

correlation exists between population and the occurrence of disasters ($\rho = 0.855$), suggesting that higher population densities

are associated with a greater frequency of disasters. Similarly, urban area size shows a strong positive correlation with both

the number of disasters ($\rho = 0.782$) and critical municipalities ($\rho = 0.753$), indicating that larger urban areas are more

susceptible to disasters, likely owing to increased population density, infrastructure complexity, and economic activities.

Additionally, demographic density exhibits a moderate positive correlation with disasters ($\rho = 0.415$), underscoring the role of

population concentration in disaster occurrence. Interestingly, the HDI moderately correlates with disaster incidence ($\rho = 0.387$), and this correlation may reflect the dual role of socioeconomic development in enhancing disaster reporting and

response capabilities while increasing the number of assets at risk.





Figure 1: Correlation analysis between disasters and critical municipalities by state, considering geopolitical, demographic, and socioeconomic variables, based on MDR, (2012); IBGE, (2021, 2022); and Brasil, (2023).

### 3.1.1 Risk assessment before 2010

Following a series of significant SRD events in the states of Rio de Janeiro (1966, 1967, 1988, 1996) and São Paulo (1967), as documented by De Ploey and Cruz (1979), Schuster et al. (2002), Vieira and Fernandes (2004), and Avelar et al. (2013), a





variety of regional methodologies gained prominence in the 1990s (Pimentel et al., 2020). By the mid-2000s, Brazil began developing systematic strategies for managing its impacts (Giustina, 2019). The government opted for a national-level unified risk assessment approach. In 2003, the first nationwide survey was started, and the PMRR was designed (Brasil, 2006) to

promote strategic planning instruments to assess risks and propose structural solutions for mitigation (Table 1). PMRR involves several stages, including risk mapping, budget estimation, intervention prioritization, and the development of alternative action matrices (Alheiros, 2006). The federal framework for conducting PMRR assessments relies on the Union Resources Decentralized Execution Agreement, which typically involves universities, public entities, or private enterprises.

### 3.1.2 Mega-disaster of the mountainous region of Rio de Janeiro in 2011

In January 2011, Brazil experienced the most catastrophic disaster in its recorded history, namely, the Mega-disaster of the mountainous region of Rio de Janeiro (Dourado et al., 2012). This calamity was induced by torrential rainfall (241.8 mm / 24 h; peak intensity of 61.8 mm / 1 h). The total rainfall recorded between January 1 and 12 was 573.6 mm (Netto et al., 2013). The impact was a series of destructive debris flows, mudflows, and a substantial number of landslides, approximately 3600 (Fonseca et al., 2021). The official records indicate 947 fatalities, more than 400 missing people, and over 50,000 left homeless

(Dourado et al., op. cit.). The financial cost was approximately US$2.5 billion (World Bank, 2012).

After this event, the government promptly implemented drastic measures to tackle sediment disasters. In the same year, Brazil established the National Plan for Risk Management and Response to Natural Disasters (MDR, 2012). The initial step involved identifying 286 municipalities as critical and 821 as vulnerable. This identification was based on the National Center for Risk and Disaster Management database and was obtained by considering historical data on public calamity decrees, states of

emergency, and human fatalities.

Additionally, the government endorsed Law No. 12.608/2012 (Brasil, 2012), establishing the National Policy of Protection and Civil Defense, National System of Protection and Civil Defense, National Council of Protection and Civil Defense, National Plan for Risk Management and Response to Natural Disasters (NPRMRND), and National Center for Monitoring and Alerts of Natural Disasters. This law represents Brazil's most significant legislative mechanism, encompassing multiple

policies to reduce the impact of socioeconomic disasters.

As an NPRMRND component, the GRS was initiated in 2011 primarily to assess high and very-high-risk zones rapidly in urban areas across those 821 vulnerable municipalities (SGB, 2023a). According to Pozzobon et al. (2018), it aims to facilitate monitoring and alert issuance (Table 1). In addition, several municipal and state entities conduct local assessments, for example, the Institute of Technological Research (IPT) in São Paulo State and the Geotechnical Institute Foundation (GeoRio)

in Rio de Janeiro Municipality, among others. However, the Federal Government's Civil House summoned the Geological Survey of Brazil, an institution under the Ministry of Mines and Energy, to assess the GRS across the country.

**Table 1: Summary of the key findings of the Brazilian surveys for disaster prevention.**



| Survey | Year | Objectives | Surveyed municipalities (December 2023) | Phenomena analyzed | Assessing topographic unit | Classes | Zone identification | Information provided | Main reference |
|---|---|---|---|---|---|---|---|---|---|
| Municipal Disaster Risk Reduction Plan (PMRR) | 2003 | Structural countermeasures planning | 729 | Landslides Flash floods Gradual floods | Hillslope, plot | 2 or 4 | Initiation | Risk areas; Affected infrastructure; Affected population; Cost-benefits; Structural countermeasures prioritization | Alheiros (2006); IBGE (2020); Mendonça et al. (2023) |
| Geological Risk Survey (GRS) | 2011 | Monitoring for early-warning, alert emissions, and emergency planning | 1676 (+277 updates) | Landslides Soil creep Rock falls Debris flows Gradual floods Flash floods | Hillslope, plot | 2 | Initiation | High and very high risk areas; Affected infrastructure; Affected population | Brasil (2007); Lana et al. (2021); SGB (2023a) |
| Susceptibility Survey (SS) | 2012 | Urban planning | 654 | Landslides Debris flows Gradual floods Flash floods | Catchment | 3 | Initiation | Susceptibility areas | Bittar (2014); Antonelli et al. (2020); SGB (2023b) |
| Geotechnical Aptitude for Urbanization Charts (GAUC) | 2014 | Urban planning | 17 | Landslides Soil creep Rockfalls Debris flows Gradual floods Flash floods | Catchment | 3 | Initiation | Different land uses aptitude classes depending on the terrain characteristics | Antonelli et al. (2021); SGB (2023c) |
| Geological Hazard Survey (GHS) | 2018 | Comprehensive Disaster Risk Reduction Management | 12 | Landslides Debris flows Rockfalls | Catchment, hillslope, plot | 4 | Initiation Transportation Deposition | Hazardous areas phenomena's maximum extent; Magnitude | Pimentel and Dutra (2018); SGB (2023d) |

### 3.1.3 Last decade

Another NPRMRND component was the launch of two additional surveys focusing on urban planning. These surveys aimed to develop a progressive assessment framework and enhance preventive strategies, mainly providing strategic information for the municipality's decision-makers to optimize land-use management. Thus, in 2012, the SS was developed to evaluate the natural predisposition of a given terrain to trigger geohydrological phenomena for urban policies (Table 1) (SGB, 2023b). In 2014, the GAUC was launched, focusing on assessing the capacity of the terrain to support various types of land use and assist

in urban development planning (Table 1) (SGB, 2023c). Between 2013 and 2017, the governments of Brazil and Japan engaged in a cooperative agreement (GIDES project). This bilateral collaboration culminated in GHS development in 2018. The GHS method is based on statistical analyses of historical events to establish topographical parameters for identifying hazardous areas and defining the phenomenon's potential maximum extent. It was designed to address some limitations of earlier surveys by diminishing the subjectivity of previous qualitative methods in categorization and estimating the trajectory and runout

extent. Furthermore, the GHS was explicitly designed to address the diverse requirements of multiple stakeholders, thereby



enabling a cohesive approach to urban planning, disaster management, and response strategies concerning the affected area (Pimentel et al., 2020).

### 3.1.4 Correlation analysis of surveys with geopolitical and socioeconomic variables

Figure 2 shows a strong-to-moderate correlation between these surveys and geopolitical, demographic, and socioeconomic
variables. The surveys showed strong positive correlations with the occurrence of disasters (PMRR, $\rho = 0.896$; GRS, $\rho = 0.686$; SS, $\rho = 0.759$) and critical municipalities (PMRR, $\rho = 0.792$; GRS, $\rho = 0.749$; SS, $\rho = 0.691$), indicating that these surveys effectively target municipalities with higher disaster incidences and heightened vulnerability. Population size (PMRR, $\rho = 0.872$; GRS, $\rho = 0.646$; SS, $\rho = 0.624$) and urban area size (PMRR, $\rho = 0.841$; GRS, $\rho = 0.582$; SS, $\rho = 0.613$) also play significant roles, suggesting prioritization of larger, more densely populated urban areas.

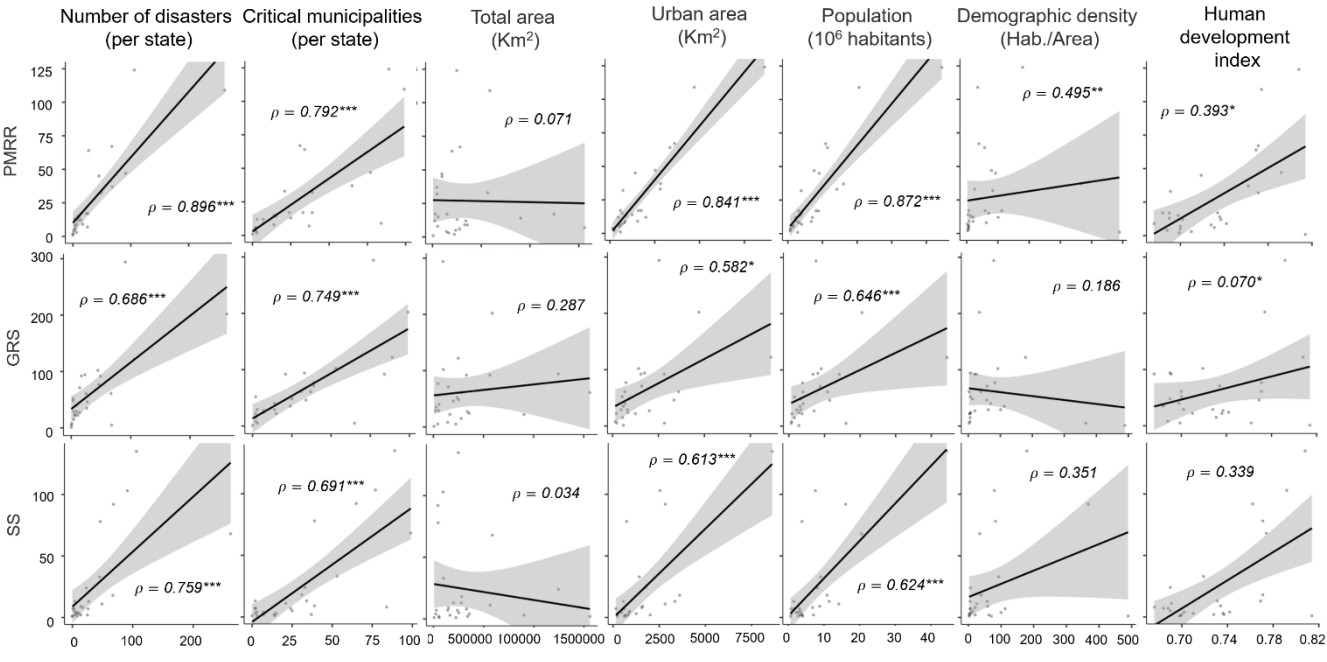

Spearman's coefficient ($\rho$). Asterisks denote significant correlations: * p value <.05, ** p value <.01, *** p value <.001. The gray area represents the 95% confidence interval.


**Figure 2: Correlation analysis of geopolitical, demographic, and socioeconomic variables with DRR assessment surveys. Analysis based on IBGE (2020) and SGB (2023a, b).**

### 3.2 Risk assessment methodologies

### 3.2.1 General description

The Geological Survey of Brazil conducts GRS, SS, GAUC, and GHS surveys, employing standardized and unified methods throughout the entire national territory (Pozzobon et al., 2018). Although PMRRs adhere to federal basic guidelines, the method varies considerably depending on local specifics and survey entities (Mendonça et al., 2023).



All methods employ heuristic-based analyses to identify hillslopes and catchments where landslides and debris flows are likely
to occur and qualify as prone areas (Table 2). The GRS is purely qualitative and relies only on expert judgment to distinguish
between high and very high classes (Lana et al., 2021). The other surveys provide probabilistic evaluations through empirical
approaches. For instance, SS uses the relief unit's landslide density index. This approach helps distinguish and categorize areas
into high or moderate classes based on their landslide density (Antonelli et al., 2020). The GAUC prioritizes moderate-to-low
susceptibility areas for evaluation, deliberately excluding areas classified as highly susceptible. Environmental protected areas
are also omitted. The characterization of geotechnical units is defined by a combination of the underlying lithological substrate
and the overlying unconsolidated covering data (SGB, 2023c). GHS applies an empirical approach based on statistical analyses
of the damage caused by past events (SGB, 2023d). Furthermore, none of these methods relies on deterministic analyses.

**Table 2: Comparative analysis of survey methods in Brazil. Analysis based on Alheiros (2006); Lana et al. (2021); Antonelli et al.
(2020; 2021); and Pimentel and Dutra (2018).**

| Criteria | | | | Survey | | | | |
|---|---|---|---|---|---|---|---|---|
| | | | | PMRR | GRS | SS | GAUC | GHS |
| Assessment method | | Unified methodology | | - | ✓ | ✓ | ✓ | ✓ |
| | Heuristic | Terrain instabilities features | | ✓ | ✓ | ✓ | ✓ | ✓ |
| | Empirical | Data-driven methods | | ✓ | - | ✓ | ✓ | ✓ |
| | Probabilistic | Field measurements | | ! | - | ! | ✓ | ✓ |
| | | Laboratory analysis | | - | - | - | ✓ | - |
| | Deterministic | Analytical methods | | - | - | - | - | - |
| | | Numerical methods | | - | - | - | - | - |
| | Analysis scale | Plot scale | | ! | ! | - | - | ! |
| | | Hillslope scale | | ! | ! | ! | ! | ! |
| | | Catchment scale | | - | - | ✓ | ✓ | ✓ |
| | Data analysis | Error | | - | - | - | - | - |
| | | Accuracy | | - | - | - | - | - |
| | | Uncertainty | | - | - | - | - | - |
| | | Precision | | - | - | - | - | - |
| | Quantitative Risk Analysis<br><br>Suggested by:<br>Fell et al. (2008) and Corominas et al. (2014) | Hazard identification prone areas | | ✓ | ✓ | ✓ | ✓ | ✓ |
| | | Hazard assessment | Frequency | - | - | - | - | - |
| | | | Return Period | - | - | - | - | - |
| | | | Magnitude | - | - | - | - | ! |
| | | | Runout analysis | - | - | - | - | ! |
| | | Exposure analysis | Temporal | - | - | - | - | - |
| | | | Spatial | ! | ! | - | - | - |
| | | Vulnerability assessment | Physical vulnerability | ! | - | - | - | - |
| | | | Social vulnerability | - | - | - | - | - |
| | | Risk estimation | Risk scenarios | ! | - | - | - | - |
| | | Quantitative assessment | Risk curves analysis | - | - | - | - | - |
| | | Qualitative assessment | Spatial multi-criteria evaluation | ! | - | - | - | - |
| | | Risk information | | ! | ! | - | - | - |
| Legend | | | | | | | | |
| Integral (✓) | Signifies complete adherence to the criteria as outlined in the literature's best practices. | | | | | | | |
| Limited (!) | Indicates partial compliance with the criteria as outlined in the literature's best practices. | | | | | | | |
| Not applicable (-) | Indicates that the method does not encompass the criteria or deviates from the best practices outlined in the literature. | | | | | | | |

The GRS integrates a heuristic approach for exposure analysis and risk area delimitation (Lana et al., 2021). On the contrary,
Mendonça et al. (2023) noted that only a limited number of PMRRs partially fulfilled the criteria for exposure analysis, physical
vulnerability assessment, risk estimation, multi-criteria spatial evaluation, and risk information. For example, despite being
advocated in the guidelines, social participation does not occur effectively. The authors emphasized that only 6% of PMRRs
adopted multi-criteria analyses. Merely 4% considered cost–benefit ratios as a prioritization criterion for structural measures.
Usually, prioritization focuses on risk classes and the number of households at risk.



The topographic units used to assess risk vary depending on the purpose of each survey. The SS, GAUC, and GHS conduct catchment analyses (Table 1; Table 2). In some instances, GHS also conducts plot scale analyses, whereas the PMRR and GRS involve partial plots analyses and hillslope examinations.

None of these surveys assesses the correlation between landslide incidents and rainfall patterns. Data analysis was conspicuously absent in all the surveys (Table 2). In addition, all the surveys considered hazard identification-prone areas. Notably, none of the surveys consider hazard assessment criteria such as frequency and return period, except for the GHS, which incorporates limited analysis of magnitude and runout distance (Pimentel and Dutra, 2018).

### 3.2.2 Input data and data collection

The surveys mainly relied on the inventory of past disasters and topographic input data (Table 3). For instance, all surveys considered the role of slope gradients (e.g., usually $\geq 25°$ for shallow landslides) (e.g., Pimentel and Dutra, 2018). Other topographical parameters are not included, except for GHS, which considers the roles of elevation, slope length, shape, and direction. (e.g., slope height $\geq 5$ m for landslide-prone areas). Usually, landslide inventory and topographic parameter data are collected through field inspection or aerial orthophotos and satellite imagery interpretations (SGB, 2023a). Recently, an emerging trend has involved the adoption of drones and unmanned aerial vehicles to acquire high-resolution imagery and updated topographic data (SGB, 2023b).

Various additional data are used for each risk assessment. The PMRR and GRS heuristically incorporate land-use and human-impact variables, such as irrigation and drainage networks, cut-fill, and dams (Table 3). These surveys use census data, such as population and building materials type (i.e., masonry and wood), to correct data about land-use and human impacts (Mendonça et al., 2023; SGB, 2023a). The SS, GAUC, and GHS partially incorporated controls related to subsurface factors, such as geological data and soil characteristics (Table 3). To assess disaster risk, these surveys used spatial correlations between phenomena, i.e., landslides, debris flow occurrences, lithology, and soil types (Pimentel et al., 2020). The SS establishes a correlation between the occurrence of geological discontinuities, such as faults or fractures (discontinuity density index), and the frequency of landslides in a given area. This approach enables a more accurate categorization (Antonelli et al., 2020). The GAUC evaluates soil properties using a combination of field and laboratory measurements (SGB, 2023c). Field tests involve tactile-visual examination and determining the soil's bearing capacity under vertical loading. The laboratory tests conducted include consistency limits (or Atterberg limits) tests to determine the soil's liquid, plastic, and shrinkage limits and particle size distribution (granulometry) tests (Antonelli et al., 2021). In contrast, the GHS utilizes measurements of soil thickness to define the maximum extent of potential phenomena (Pimentel and Dutra, 2018). However, none of these surveys considered triggering factors, such as rainfall, precipitation, or thresholds, for assessing critical areas.

**Table 3: Comparative analysis of the input data per survey in Brazil. Analysis based on Alheiros (2006); Lana et al. (2021); Antonelli et al. (2020; 2021); and Pimentel and Dutra (2018).**



| Criteria | | | | Survey | | | | |
|---|---|---|---|---|---|---|---|---|
| | | | | PMRR | GRS | SS | GAUC | GHS |
| Input data | Preparatory factors | Surface condition | Inventory | ✓ | ! | ✓ | ✓ | ✓ |
| | | Topography | Elevation | - | - | - | - | ✓ |
| | | | Slope gradient | ✓ | ✓ | ✓ | ✓ | ✓ |
| | | | Slope direction | - | - | - | - | ! |
| | | | Slope length | - | - | - | - | ✓ |
| | | | Slope shape | - | - | - | - | ✓ |
| | | | Stream network and drainage density | - | - | ! | - | - |
| | | Land use and anthropogenic factors | Buildings | ! | ! | - | - | - |
| | | | Drainage and irrigation networks | ! | ! | - | - | - |
| | | | Dams and reservoirs | ! | ! | - | - | - |
| | | | Vegetation | ! | ! | ! | ! | ! |
| | | | Informal human interventions | ! | ! | - | ! | ! |
| | Underground condition | Geology | Lithology | - | - | ! | ✓ | ! |
| | | | Structural | - | - | ! | ✓ | ! |
| | | | Discontinuities | - | - | ! | ✓ | ! |
| | | Soil | Soil type | - | - | ✓ | ✓ | ✓ |
| | | | Soil thickness | - | - | - | ✓ | ✓ |
| | | | Soil strength | - | - | - | ✓ | - |
| | | | Bedrock strength | - | - | - | - | - |
| | | | Geotechnical properties | - | - | - | ✓ | - |
| | | | Soil/bedrock hydraulic parameters | - | - | - | ! | - |
| | | | Groundwater | - | - | - | ! | - |
| | Triggering factor | | Rainfall analysis | - | - | - | - | - |
| | | | Magnitude/Frequency relations | - | - | - | - | - |
| | | | IDF curves | - | - | ! | - | - |
| | | | Rainfall thresholds | - | - | - | - | - |
| | | | Temperature | - | - | - | - | - |
| | | | Humidity | - | - | - | - | - |
| Legend | | | | | | | | |
| Integral (✓) | Signifies complete adherence to the criteria as outlined in the literature's best practices. | | | | | | | |
| Limited (!) | Indicates partial compliance with the criteria as outlined in the literature's best practices. | | | | | | | |
| Not applicable (-) | Indicates that the methodology does not encompass the criteria or it deviates from the best practices outlined in the literature. | | | | | | | |

### 3.2.3 Outcomes and output data

The information provided depends on the objective of the investigation. Furthermore, except for the PMMR, all other survey results are readily accessible online (SGB, 2023a, b, c, d). Generally, these datasets encompass technical reports (except for SS), thematic charts, and GIS data. Most surveys classify sensitive areas into 2–4 levels, except for GAUC, wherein the number of classes varies depending on the municipality's complexity (Table 1).

All these surveys provide data concerning sediment-related phenomena, including landslides, debris flow, and rockfall (Table

4). All surveys also identified the potential initiation zone. The GHS also provided a potential area for sediment transport and deposition. The first two surveys, PMRR and GRS, aim to support structural and nonstructural countermeasures planning (Table 1), providing information regarding the impact on infrastructure and affected populations. Specifically, PMRR offers an inventory of SC tailored to mitigate risks and cost–benefit estimations.

Moreover, SS provides a legend with detailed descriptions and illustrative representations of distinct classes explicitly tailored

to urban planning. The GAUC survey categorizes terrain into three primary aptitude classes (high, moderate, and low), which are subdivided into specific geotechnical units, offering detailed insights and recommendations for each unit's land-use planning. It also presents supplementary maps and strongly emphasizes information related to aptitude classes and soil characteristics. GHS focuses on topographic data, providing hazardous-prone areas, delineating potential maximum extents, and distinguishing them into critical and dispersion areas (Table 4).





**Table 4: Comparative analysis of the output data per survey in Brazil. Analysis based on Alheiros (2006); Lana et al. (2021); Antonelli et al. (2020; 2021); and Pimentel and Dutra (2018).**

| Criteria | | | Survey | | | | |
|---|---|---|---|---|---|---|---|
| | | | PMRR | GRS | SS | GAUC | GHS |
| Output data | Physical characterizer of soil movement | Type of Movement | ✓ | ✓ | ✓ | ✓ | ✓ |
| | | Speed of Movement | - | - | - | - | - |
| | | Moisture Content | - | - | - | - | - |
| | | Grain Size and Distribution | - | - | - | ! | - |
| | | Soil Structure | - | - | - | ! | - |
| | | Slope Angle and Topography | - | - | - | - | ✓ |
| | | Vegetation Cover | - | - | - | - | - |
| | | Human Activity | ! | ! | - | - | - |
| | | Shear Strength | - | - | - | ! | - |
| | | Load Capacity | - | - | - | ✓ | - |
| | | Erosion Rate | - | - | - | ! | - |
| | | Land Use History | - | - | - | - | - |
| | Zone identification | Initiation zone identification | ✓ | ✓ | ✓ | ✓ | ✓ |
| | | Transportation zone | - | - | - | - | ✓ |
| | | Deposition zone | - | - | - | - | ✓ |
| | Impact analysis | Affected infrastructure | ✓ | ! | - | - | - |
| | | Affected population | ✓ | ! | - | - | - |
| | | Cost-benefits | ! | - | - | - | - |
| | | Frequency | - | - | - | - | - |
| | | Magnitude | - | - | - | - | ! |
| Legend | | | | | | | |
| Integral (✓) | Signifies complete adherence to the criteria as outlined in the literature's best practices. | | | | | | |
| Limited (!) | Indicates partial compliance with the criteria as outlined in the literature's best practices. | | | | | | |
| Not applicable (-) | Indicates that the methodology does not encompass the criteria or it deviates from the best practices outlined in the literature. | | | | | | |

In addition, Rocha et al. (2018) evaluated the effectiveness of the SS, GHS, and GAUC methods in the municipality of Nova Friburgo by assessing the landslides resulting from the 2011 mega-event in the mountainous region of Rio de Janeiro. The

study found that the GHS method outperformed the others, achieving an accuracy coefficient of 95% in identifying areas destroyed by the disaster. Conversely, the SS and GAUC methods exhibited 55% accuracy and were significantly less effective in detecting the affected areas.

# 4 Evaluation Results of Risk Assessment Surveys

## 4.1 Implemented status of surveys

These surveys are still in progress in 2024 and have not yet achieved nationwide coverage. Based on information from IBGE (2020), the PMRR was carried out in 729 municipalities (Fig. 3a). However, a comprehensive national repository for these data is currently unavailable. Espírito Santo and Rio de Janeiro are the only states with around 40% of their municipalities surveyed. In Amapá and Pernambuco, surveying has been done for approximately one-quarter of their municipalities. The remaining states have less than 20% of their municipalities surveyed.



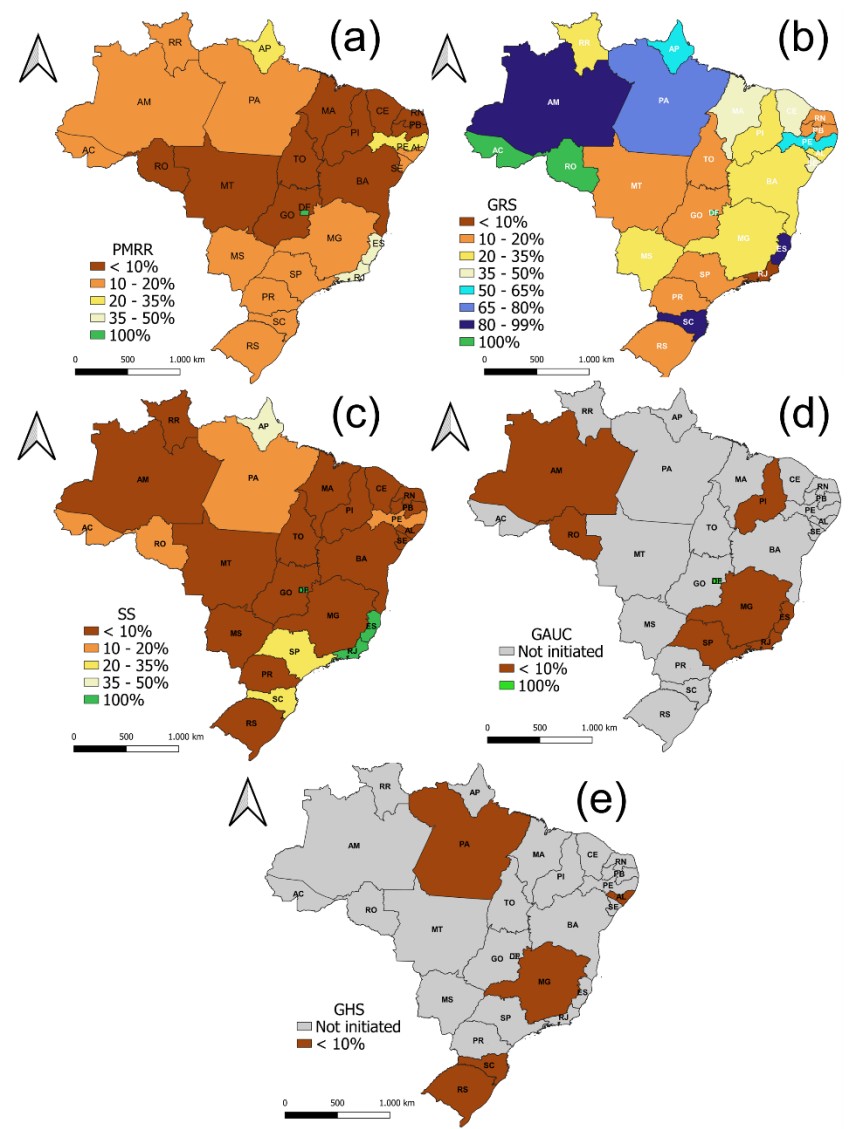

**Figure 3: Coverage of state-level DRR surveys. Analysis based on IBGE (2020) and SGB (2023a, b, c, d). The current status refers to December 2023.**

The GRS was surveyed in 1676 municipalities and implemented updates for 277 (Table 01; Fig. 3b). According to the SGB (2023a), Acre, Rondônia, and Distrito Federal have been thoroughly surveyed. Amazonas, Santa Catarina, and Espírito Santo covered over 90% of the state. Several states have achieved significant progress, covering over 50% of their territory, notably in Amapá, Pará, Pernambuco, and Roraima. In contrast, other states are still in the initial phases of surveying, with less than 20% coverage, including Goiás, Minas Gerais, Paraíba, Paraná, Piauí, Rio de Janeiro, Rio Grande do Norte, Rio Grande do Sul, São Paulo, Sergipe, and Tocantins. Notably, the Department of Mineral Resources of the State of Rio de Janeiro (DRM-




RJ) surveyed 87% of the state using a different methodology for assessing imminent geological risk, which was not evaluated
in this study (DRM, 2023).

The number of municipalities surveyed for SS is relatively like PMRR. According to the SGB (2023b), the surveying processes
in Distrito Federal, Espírito Santo, and Rio de Janeiro have been completed, followed by Amapá (43%), Santa Catarina (34%),
and São Paulo (20%), which are at more advanced stages of survey completion (Fig. 3c). Conversely, the remaining states
demonstrate less than 20% coverage. Furthermore, more recent surveys, such as GAUC and GHS, have assessed only 17 and
12 municipalities, respectively (SGB, 2023c, d). GAUC's efforts are primarily concentrated in Rio de Janeiro state, achieving
5% of coverage (Fig. 3d). In contrast, GHS is focused on Santa Catarina, with 2% coverage (Fig. 3e).

## 4.2 Examining the correlation between risk assessment surveys and the implementation of disaster reduction countermeasures

Risk assessment surveys are vital resources for various risk-management initiatives. Therefore, the effectiveness of these
surveys can be evaluated by examining the activities and initiatives developed from the basic information provided by them.
Figure 7 illustrates the distribution of municipalities across the Brazilian states that have adopted various DRR initiatives, such
as master plans (MP), landslide–specific laws (SL), land-use and land-occupation laws (LULOL), early warning systems
(EWS), emergency response plans (ERP), and structural countermeasures (SC). As indicated, only 53% of municipalities have
established MPs. Within this group, just 25% of the MPs incorporate specific instruments for sediment-related disaster
reduction. The data also reveal significant disparities in adopting MPs for urban planning across states (IBGE, 2020). For
example, a high percentage of municipalities in states like Paraná (99%), Pará (83%), Mato Grosso (81%), and Santa Catarina
(80%) have implemented MPs. In contrast, states such as Paraíba (26%), Rio Grande do Norte (26%), Piauí (32%), Sergipe
(35%), and Minas Gerais (38%) show considerably lower implementation rates.

A deeper analysis of the MPs reveals the extent to which they incorporate measures for mitigating sediment-related disasters.
The Federal District and states like Rio de Janeiro (42%), Acre (40%), Espírito Santo (39%), and Pernambuco (37%) lead in
integrating such strategies. Conversely, states including Tocantins, Rondônia, Amazonas, Amapá, and others show less than
20% incorporation of these preventive measures in their MPs, highlighting regional disparities in DRR emphasis (Fig. 4).

The northern region shows particular deficiencies in disaster prevention, with states like Roraima and Rondônia lagging
significantly; 80% and 77% of their municipalities, respectively, lack additional disaster prevention strategies. In the Northeast,
disparities are stark, with low implementation rates in states like Piauí, Rio Grande do Norte, and Paraíba. Pernambuco stands
out in terms of more comprehensive DRR efforts. In contrast, the Southeastern region displays the highest DRR engagement
in Brazil, with Minas Gerais mapping 30% of risk areas but only 5% implementing protective measures. Espírito Santo and
Rio de Janeiro demonstrate higher implementation rates in various DRR strategies, whereas São Paulo faces challenges, with
54% of its municipalities not implementing any measure (Fig. 4).





**Figure 4: Distribution of municipalities in Brazilian states implementing sediment-related DRR initiatives (in percentages). Analysis based on IBGE (2020) DRR dataset.**

Santa Catarina leads the Southern region in disaster risk mapping, yet lags in engineering and EWS implementations. Paraná shows moderate DRR implementation, and Rio Grande do Sul has the lowest engagement. The Distrito Federal excels in DRR strategies in the Middle-Western region, although it lacks engineering plans. The DRR efforts in Mato Grosso and Goiás are inadequate, and Mato Grosso do Sul shows a mixed performance, emphasizing the regional challenges in DRR strategy implementation (Fig. 4).



The effectiveness of these surveys was evaluated using Spearman's rank correlation coefficient to assess the association between adopting specific DRR strategies across 5570 municipalities and the outcomes from the corresponding risk assessment
surveys (Fig. 5). Because the GAUC and GHS have low implementation levels, they were removed from this analysis.

The PMRR survey showed a remarkably strong correlation with most DRR initiatives, indicating a robust interconnection between comprehensive DRR strategies and PMRR. The strongest correlation was observed with the MP ($\rho = 0.927$, p <.001), SC ($\rho = 0.922$, p <.001), ERP ($\rho = 0.901$, p <.001), and EWSs ($\rho = 0.843$, p <.001). In addition, GRS shows a significant correlation with SC ($\rho = 0.710$, p <.001), ERP ($\rho = 0.649$, p <.001), MP ($\rho = 0.642$, p <.001), and LULOL ($\rho = 0.635$, p <.001)
and moderate to others initiatives. Further, SS also shows a strong correlation with SC ($\rho = 0.812$, p <.001), ERP ($\rho = 0.674$, p <.001), and EWS ($\rho = 0.742$, p <.001). It has a moderately high correlation with other initiatives.

The positive correlation between surveys and DRR strategies indicates that adopting risk assessment surveys enhances municipalities' implementation of comprehensive DRR strategies. The McDonald's Omega ($\omega$) analysis, yielding a value of 0.964, was conducted to ensure the reliability of the results. It demonstrated excellent internal consistency, thereby supporting
the validity of our Spearman correlation findings and ensuring the robustness and credibility of the collected data.

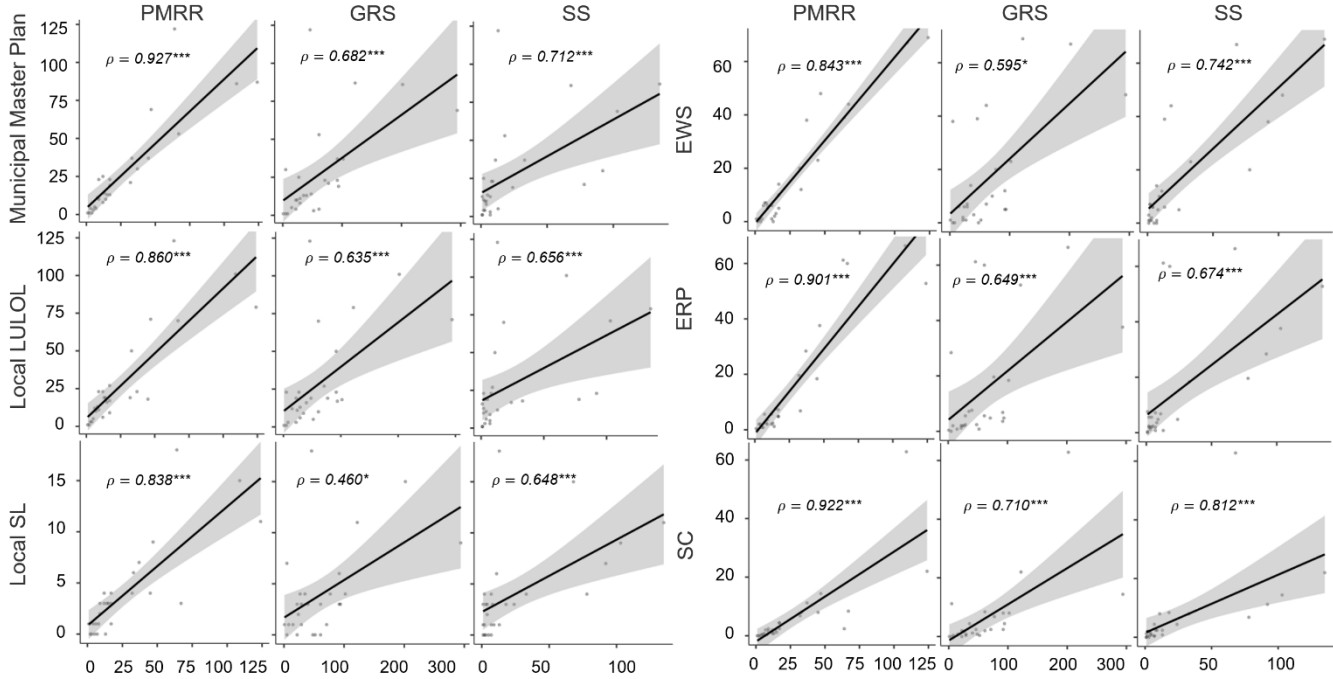

Spearman's coefficient ($\rho$). Asterisks denote significant correlations: * p value <.05, ** p value <.01, *** p value <.001. The gray area represents the 95% confidence interval.

**Figure 5: Spearman correlation displaying the relationship between the most important DRR initiatives and risk assessment surveys across 5570 municipalities. Analysis based on IBGE (2020) and SGB (2023a, b).**




## 4.3 Applicability for disaster reduction initiatives

A relevance matrix (Fig. 6) can help to understand the applicable range of these surveys (Hungr et al., 2005; Fell et al., 2008; Corominas et al., 2014). The color gradient, ranging from dark blue (0—Not applicable) to dark red (4—Fully applicable), indicates the level of applicability of each element according to the survey scales.

The PMMR and GRS can be applied to various initiatives, existing urban developments, enhancing EWSs, formulating emergency plans, and disseminating practical information (Fig. 6). However, their application is less prevalent in urban
planning. Furthermore, while they may be used to support local legislative frameworks, they are not recommended for supporting national legislative policies.

The SS and GAUC can apply for similar initiatives, such as information communication and dissemination, urban planning and development, enhancing warning systems, and implementing climate change countermeasures, but they are not commonly employed in developing emergency plans. Although the SS is not applicable for prioritizing SC, it is highly relevant to the
legislative framework at both the national and municipal levels. In contrast, the GAUC is not commonly used for national legislative policies. GHS achieves the widest applicability. This survey can apply all initiatives shown in Fig. 6, although some are partially applicable, especially in framing national policies.

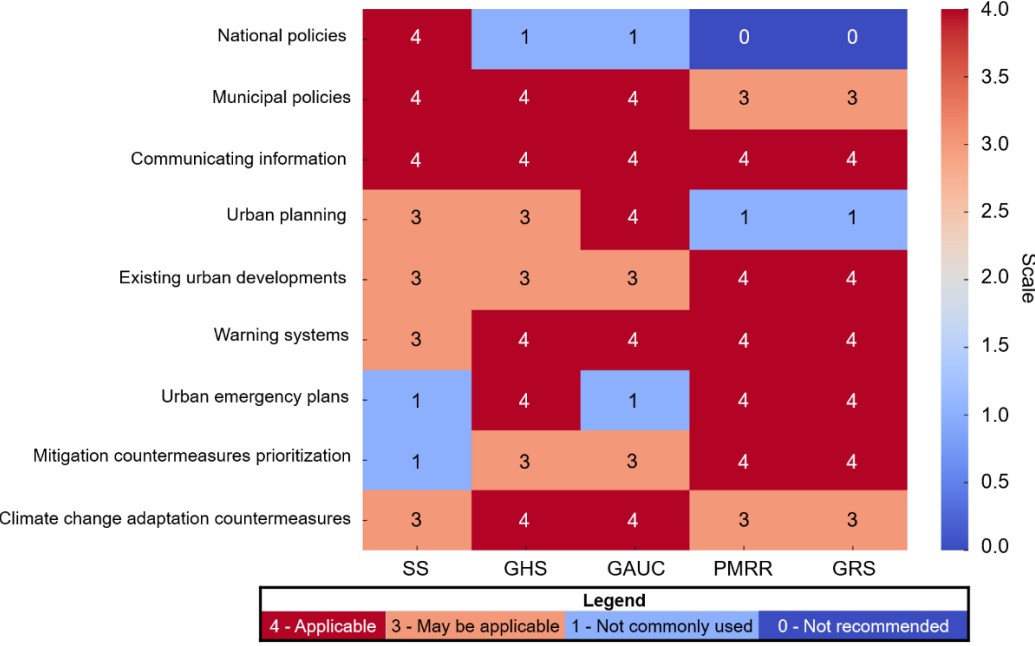

**Figure 6: Brazilian DRR assessment relevance matrix based on applicability recommendations from Hungr et al. (2005), Fell et al.**
**(2008), and Corominas et al. (2014).**

## 4.4 Cost–beneficiary analysis

Table 5 provides an overview of Brazilian surveys implemented in a medium-sized municipality, with notable variations in cost–beneficiary ratios. The PMMR survey exhibits a high cost per beneficiary of $6.66, indicating inefficiencies that may



benefit from an updated review. The initial phase, involving mapping risk areas, requires at least eight months to complete
(IPPLAN, 2016). Despite the lack of public data availability for subsequent phases, the study was reported to exceed one year. According to Mendonça et al. (2023), the duration and costs vary by municipality, indicating a flexible approach that accounts for territorial extent and inherent complexity. Conversely, the GRS method reached the second-highest position in the evaluation. The GRS, which is faster and more cost-effective than PMRR, achieved a cost–benefit ratio of 1.309, which showcased its economic and practical approach.

The SS achieved the highest position, securing the maximum average number of beneficiaries per municipality owing to its low average runtime and pricing, as detailed in Table 5. In addition, this survey stood out with an exceptionally low cost of $0.0004 per beneficiary, demonstrating highly effective resource use. Meanwhile, the GAUC and GHS surveys showed mixed results. The GAUC showed higher costs, longer execution durations, and a cost–beneficiary ratio of 0.1429. Notably, despite being established four years before the GHS (Table 1), the GAUC shows a smaller average number of beneficiaries. The GHS

features a reasonable average implementation time and costs, indicated by a reasonable cost of $0.36 per beneficiary holding the third position.

**Table 5: Cost per beneficiary's attributes of Brazilian DRR surveys.**

| Survey | Runtime (Months) | Pricing (US$) | Total Beneficiaries | Average Beneficiaries per Municipality | Cost per Beneficiary |
|--------|------------------|---------------|---------------------|----------------------------------------|----------------------|
| PMRR | 12 | 20,000 | *Not specified* | 3,000 | 6.667 |
| GRS | 1 | 3,200 | 4,000,000 | 2,445 | 1.309 |
| SS | 2 | 6,500 | 100,000,000 | 757,137 | 0.009 |
| GAUC | 7 | 32,000 | 450,000 | 28,125 | 1.138 |
| GHS | 5 | 18,000 | 500,000 | 50,000 | 0.360 |

## 5 Discussion

Based on the results, we structured the discussion into three key questions: 1) Are any elements lacking in the current landslide
risk assessments? Referencing Sect 3, we debate these techniques and their feasibility, identifying deficiencies and indicating potential enhancements based on global best practices. 2) What is the reason for the low implementation rate of risk assessments? Drawing from insights in Sect 3 and 4 and literature references, we discuss the existing DRR framework and examine the potential reasons behind Brazil's low implementation rate of these assessments. 3) Does risk assessment effectively contribute to initiatives related to sediment disaster prevention? Based on the findings in Sect 3 and 4, we explore

the effectiveness of risk assessment surveys in contributing to sediment disaster prevention initiatives.





**5.1 Are there any elements lacking in the current sediment-related risk assessment surveys?**

**5.1.1 Specific deficiencies and recommendations**

Current risk assessment methodologies in Brazil exhibit substantial limitations, redundant efforts, and significant challenges (Tables 2 to 4). This underscores the urgent need for methodological improvements and the adoption of more comprehensive
quantitative approaches. The SS and GRS methods are well established and cover many surveyed municipalities. Nevertheless, they are subject to several limitations. Thorough updates and refinements may face significant implementation challenges. Despite the impressive national coverage of PMMR, an update accompanied by thorough and unified procedural guidelines is clearly needed, as demonstrated by Mendonça et al. (2023). Given the restricted coverage, the GAUC and GHS surveys were considered the most suitable for methodological improvements. Recognizing the complementary nature of the five surveys,
we propose a nuanced, systematic, and progressive multilevel approach to risk assessment that aligns with global best practices and considers local specifics (e.g., Fell et al., 2008; Corominas et al., 2014).

**– Absence of data collection and updates**

**Disaster database and data analysis**

Effective DRR management relies heavily on high-quality and consistent data. Guzzetti et al. (2012) emphasized the need for
practical working definitions and optimal inventory databases, while Guzzetti et al. (1999) highlighted the importance of uniform terminology for consistent technical databases. Fell et al. (2008) also noted that the efficacy of DRR management is deeply linked to data quality. Despite significant investments, Brazil still lacks unified and standardized disaster databases and inventories, leading to significant challenges in subsequent efforts. Therefore, as recommended by Froude and Petley (2018), UNDRR (2020), and Corominas et al. (2023), Brazil needs to adopt standardized terminology for sediment-related phenomena
and establish a unified national technical database. Guzzetti et al. (2006) also stated no consensus on validating the surveys. Brazilian assessments' current data analysis processes are solely based on heuristic subjective field inspections (Dias et al., 2021). Therefore, incorporating an analysis of the models' accuracy based on inventories of past events, as suggested by Fell et al. (2008), is highly recommended.

**Demographic data**

The PMRR and GRS methods estimate potentially affected populations and damaged areas (Table 2) based on 2010 census data (Mendonça et al., 2023). However, rapid urban sprawling leads to underestimations due to outdated information (Silva Rosa et al., 2015). Consequently, these assessments may not accurately reflect current risk stages, but represent conditions from almost 15 years ago. However, recognizing communal efforts that assess and monitor Brazil's population occupation and migration patterns is essential. To address this challenge, a semi-quantitative approach is recommended for accurate and
consistent analysis. Souza et al. (2019) combined 2010 census data with GRS survey outcomes to develop the BATER (Statistical Territorial Risk Base), providing insightful data on at-risk population exposure. Alternatively, Emberson et al.



(2020) adjusted the world-gridded population dataset with the local population density index to estimate population exposure and used OpenStreetMap to evaluate infrastructure exposure. Integrating these procedures with the 2022 demographic census data (IBGE, 2022) would enhance accuracy and provide updated population exposure estimates, particularly in the GRS

survey.

**Site-condition data**

These surveys rely heavily on topographic parameters (e.g., Reichenbach et al., 2018; Dias et al., 2021) and underground conditions (Table 3). Nevertheless, most data utilized as input typically date back at least five years ago, making it challenging to properly evaluate terrain modifications due to human activities. A coherent approach includes continuously updating

topographic data to ensure these assessments remain relevant and accurate. Due to similar challenges faced with demographic data, regular updating at the national level is complex to implement. To overcome this issue, establishing a shared database with relevant stakeholders is advisable for a more precise evaluation. For instance, Junichi and Naoki (2020) mentioned the public–private consortium as a systematic approach to leveraging advanced technologies and expertise in Japan.

The GAUC survey's accuracy in assessing soil characteristics can be enhanced by integrating global best practices in

geotechnical testing and statistical analysis. Enhancements such as standard and cone penetration tests (Uchida et al., 2009), geophysical techniques (Bortolozo et al., 2018), rock quality designation analysis (Zheng et al., 2018), and building information modeling (Khan et al., 2021) provide detailed subsurface analysis and a better understanding of soil and bedrock properties. Infiltration modeling improves water dynamics knowledge (Failache and Zuquette, 2018), and statistical methods like fuzzy clustering reduce subjectivity in geotechnical classifications (Hossein Morshedy et al., 2019).

**– Lack of output data**

**Frequency and magnitude**

Evaluations of frequency and magnitude are crucial for creating accurate quantitative risk assessments and informing decision-making processes (Corominas and Moya, 2008), particularly in regions like Brazil, where the absence of temporal context restricts the predictive accuracy of the likelihood of events. A gap that becomes increasingly problematic in the climate change

context (Marengo et al., 2021). Incorporating frequency and enhancing magnitude analyses in the GHS survey is essential for improving hazard assessment and prediction (Table 2). Temporal and spatial analyses can be performed using a variety of statistical techniques (e.g., Guzzetti et al., 1999; Guthrie and Evans, 2004; Ayalew and Yamagishi, 2005; Guzzetti et al., 2005). Incorporating these elements into the SS survey would provide more detailed information for stakeholders (Segoni et al., 2018).

**Runout area**

The runout area evaluation allows a more precise prediction of sediment-related events' trajectory, extent, and impact (Rickenmann, 2005; Corominas et al., 2014). Grasping the potential reach and force of material movement allows planners





and decision-makers to more effectively foresee an event's possible consequences and intensity, facilitating the development of customized strategies (Fell et al., 2008). Although all surveys assessed initiation-prone areas, only the GHS provided a limited understanding of runout distances (Table 2). However, its reliance on national parameters limits applicability to local contexts, underscoring the need for improvement that reflects specific regional conditions (McDougall and Hungr, 2004;
McDougall, 2017). Incorporating power-law equations and an empirical–probabilistic approach has been demonstrated to yield satisfactory outcomes, as evidenced by Corominas (1996), Rickenmann (1999), Guzzetti et al. (2002), Legros (2002), Kimura et al. (2014), and Brideau et al. (2021), Di Napoli et al. (2021).

**Socioeconomic impacts data**

Socioeconomic indicators are essential for building resilience and providing a solid foundation for adaptive mitigation strategies (Fell et al., 2008). Historically, risk analysis in Brazil has primarily focused on physical vulnerabilities (Mendonça et al., 2023), as evidenced by Sect 3.2.1. While disasters arise from the interplay between physical and societal factors (Veyret, 2007), extensive disaster research has consistently demonstrated that impacts vary significantly across different societal segments (*e.g.,* Sutley and Hamideh, 2020). Addressing this issue requires integrating GRS and PMRR vulnerability societal indicators (Wisner, 2016). These criteria should include educational levels, risk perception, demographic details such as age and the needs of special groups, and economic conditions such as income levels and local economic disparities. These parameters enable the creation of detailed thematic risk maps and tailored strategies (Miguez et al., 2018; Sims et al., 2022). For instance, Fox et al. (2024) integrated socioeconomic factors with traditional risk models to provide a more holistic vulnerability analysis. As a suggestion, the IBGE (2022) data contain numerous socioeconomic features that can be used to refine the accuracy and effectiveness of vulnerability assessments.

**5.2 What are the reasons for the low implementation rate of risk assessments?**

Figure 2 confirms the strategic approach that aims to maximize the number of beneficiaries and mitigate extensive damage in areas with larger populations and more significant infrastructure. However, despite the strong correlations identified, the surveys exhibited low implementation rates (Fig. 3). Therefore, we analyzed the potential challenges contributing to these low rates and gain insights into areas that warrant improvements to enhance national coverage.

**5.2.1 Political issues**

The results showed a strong positive correlation between disasters and urban areas ($\rho = 0.782$) as well as the population ($\rho = 0.855$), indicating a significant impact on urban populations (Fig. 1). The political landscape greatly influences the implementation of DRR strategies, especially in regions with unstable governance or chronic political instability (Veyret, 2007; Nogueira et al., 2014). Such instability often leads to a lack of sustained commitment to DRR policies, with political priorities shifting with leadership changes, resulting in inconsistent support and funding (Almeida, 2015). Resource allocation often competes with other political agendas, marginalizing DRR initiatives (Ganem, 2012). The results showed low adherence



to surveys in the MPs (14%) (Fig. 4). The strong correlation between legislative prevention mechanisms (MP, LULOL, SL) and surveys (Fig. 5) indicates a trend toward implementing legal measures in disaster-affected municipalities. However, adoption rates for local land use and occupation (LULOL) and sediment disaster prevention (SL) laws are still low, at 14% and 2%, respectively, highlighting a lack of political commitment to disaster risk management. The low engagement and discontinuity in subsequent initiatives underscore the significant impact of political dynamics on the DRR cycle, consistent with Silva's (2022) findings. Another contributing factor is public managers' general lack of awareness or underestimation (Londe et al., 2015). Henrique and Batista (2020) suggested that many officials may not fully recognize the value of these surveys in preventing sediment-related disasters, leading to their lower priority. Challenges in implementing PMRR arise from decentralizing government resources and varied execution by different entities, leading to inconsistent outcomes (Mendonça et al., 2023). Such decentralization requires active participation and financial investment from municipal administrators, thus limiting effective risk mitigation (Raposo, 2019).

### 5.2.2 Demographic issues

Brazil's demographic dynamics pose significant challenges to the effective implementation of DRR strategies. Correlating data from IBGE (2020; 2022) demonstrates a strong positive association between population size and the occurrence of disasters ($\rho = 0.855$), indicating that areas with higher population densities, especially urban areas ($\rho = 0.782$), are more prone to frequent disasters (Fig. 1). These observations align with global trends, reported by Maes et al. (2017), and Brazilian trends, highlighted by Alvalá et al. (2019). According to Ozturk et al. (2022), rapid urbanization further compounds these vulnerabilities, as more people live in hazardous-prone areas lacking sufficient infrastructure, thereby complicating the implementation and effectiveness of DRR initiatives. Urban sprawl frequently causes disjointed and unplanned development, leading to significant gaps in the coverage and accuracy of risk assessment data (Bozzolan et al., 2023). Consequently, these data often become outdated or incomplete.

### 5.2.3 Socioeconomic issues

The results demonstrated a weak positive correlation between the HDI and disaster incidence ($\rho = 0.387$) (Fig. 1). This correlation suggests a dual phenomenon: municipalities with higher socioeconomic development have better disaster reporting, DRR infrastructure, and improved capabilities (Ozturk et al., 2022). Conversely, increased economic activities and asset accumulation elevate potential losses and recovery costs. Socioeconomic disparities influence public and political engagement. Thus, municipalities with lower HDI may struggle with effective DRR strategies due to limited resources and infrastructure (Lin et al., 2023). Data show that municipalities with higher HDI scores tend to implement more PMRR (Fig. 2; $\rho = 0.393$). As PMRR implementation is driven by direct municipal demand, regions with higher socioeconomic development may likely adopt advanced disaster management solutions. This assumption corroborates findings from Londe et al. (2015), who emphasized that local capacities vary markedly depending on resource allocation and institutional support. The moderate positive correlation with SS ($\rho = 0.339$) may suggest that the Geological Survey of Brazil targets municipalities with basic



financial resources. Further, the strong correlation between SS and PMRR ($\rho = 0.76$) may indicate that municipalities with SS

surveys are likely to implement PMRR, underscoring the importance of progressive surveying. Conversely, the very weak

correlation with GRS ($\rho = 0.070$) implies that socioeconomic status minimally influences the implementation of this survey,

indicating a standardized approach across diverse HDI rates.

### 5.2.4 Specific reasons

Fell et al. (2008) observed that evaluations predominantly involve basic or intermediate assessments due to data scarcity and

cost limitations. The extent of national coverage varied significantly based on the launch year, methodological complexity

(Table 1), and financial costs (Table 5). Furthermore, the results reveal a trade-off between the costs, execution time, and

breadth. The applicability relevance matrix (Fig. 6) analyzes the aggregated significance of each survey in the DRR framework.

According to Lana et al. (2021), the GRS, launched in 2011, uses only simple field heuristic analysis to identify risk areas,

estimates vulnerable buildings with satellite imagery, and extrapolates the affected population by multiplying the number of

houses by four, the average household size in Brazil. In contrast, PMRR, established in 2004, involves a complex multistage

process involving risk mapping, proposing countermeasures, cost estimation, prioritization, and securing funding, as Alheiros

(2006) explained. The analysis indicates that the GRS ranks as the cheapest survey. Conversely, PMRR registers the highest

costs. This disparity is evident in the coverage statistics: GRS encompasses 1676 municipalities, whereas PMRR, despite being

launched eight years before GRS, covers only 729 (Table 1). Established in 2012, the SS is characterized by a simple and well-

consolidated methodological procedure (Antonelli et al., 2020) that has gained broad acceptance among practitioners and

provides an excellent cost per beneficiary. Despite these strengths, the applicability of SS is considered moderate, particularly

when compared to newer surveys (Fig. 6). In contrast, the GAUC, introduced in 2014, and the GHS, launched in 2018,

demonstrate high applicability in DRR initiatives. However, these surveys present challenges, such as moderate execution time

and inherent costs, and require more detailed analysis (Antonelli et al., 2021; Pimentel and Dutra, 2018). Due to these

complexities, they offer moderate to high costs per beneficiary and have relatively low adherence among practitioners.

### 5.3 Does risk assessment surveys effectively contribute to initiatives related to sediment disaster prevention?

The IBGE (2020) data analysis confirmed a lack of integration between surveys and urban planning, indicating a missed

opportunity for embedding proactive risk management in urban development. Brazilian municipalities with smaller

populations (up to 50,000 residents, 87.8%) face significant challenges in implementing effective DRR countermeasures, as

discussed in Sect 4.2. In contrast, municipalities with higher population densities (12.2%) generally adopt a more

comprehensive approach. Moreover, integrating these survey outcomes into municipal MPs is methodologically complex and

resource intensive, requiring specialized skills and adequate funding (Londe et al., 2023). This complexity can deter

municipalities from effectively utilizing survey data for planning (Amaral, 2019; Bonelli et al., 2022).

The data show a predominant focus on emergency response (ERP) over preventive measures (MP, LULOL, SP, EWS, EDAS)

and mitigation actions (SC), indicating a reactive stance in disaster risk governance (Fig. 4). Low adherence to risk assessment





surveys hampers the implementation of essential risk reduction countermeasures. This trend aligns with the observations by Maes et al. (2017) in other tropical countries. However, states such as Rio de Janeiro, Espírito Santo, Santa Catarina, and the Federal District, where over 80% of risk mitigation initiatives are concentrated, also exhibit the highest survey coverage. This

suggests an indirect correlation between the number of surveys conducted and the implementation of subsequent DRR initiatives. The correlation analysis (Fig. 5) confirms the importance of these surveys in enhancing sediment disaster reduction measures, particularly PMRR and SS, which highly correlate with MPs, LULOL, SC, and ERPs. McDonald's Omega analysis validates the reliability of the results, reinforcing the foundational role of these surveys in DRR initiatives.

## 6 Conclusions

Brazil's extensive DRR efforts reflect a robust approach to managing sediment-related hazards. The country employs five national risk assessment surveys: PMRR, GRS, SS, GAUC, and GHS. These diverse methodologies collectively enhance the understanding of geohazards and support subsequent disaster management strategies. Our analysis confirms that these surveys improve sediment disaster management by enabling targeted, evidence-based strategies. However, prevention strategies are critically under-implemented, particularly in smaller municipalities. Despite legal mandates, 87% of municipalities have not

integrated DRR into their MPs; this discrepancy reveals a gap between legislative intent and administrative practice, highlighting the need for more effective and comprehensive policies. While significant progress has been made on these surveys, challenges, including achieving expressive national coverage, improving data collection, enhancing output information, and incorporating cutting-edge technologies, remain. A standardized, synergistic framework is essential for effective risk management, and essential recommendations include developing a unified technical database, refining survey

methodologies, enhancing data collection, and focusing on uncertainty analysis. Adopting probabilistic models and leveraging data analytics will strengthen management capabilities. Additionally, analysis of event frequency, magnitude, rainfall thresholds, and physical and socioeconomic vulnerability assessment integration are crucial. The analysis of applicability and cost-beneficiaries revealed that the GRS and SS surveys were the most cost-effective despite their restricted applications. Furthermore, PMRR encounters considerable challenges related to reproducibility, costs, and execution time. Though unique,

the GAUC is costly and less representative nationally, requiring methodological improvements before widespread adoption. GHS excels in delimiting runout hazardous areas and provides crucial information for planning and response, especially for climate change adaptation. Despite its balanced cost per beneficiary, its limited representativeness requires legislative endorsement and methodological refinement to ensure practical application in DRR strategies. Enhancing these methods will improve Brazil's resilience to potential hazards and readiness for climate change impacts. Finally, given the complementary

nature of these surveys, we recommend an integrated, progressive, and strategic approach.



**Author contribution**

TS was responsible for initial conceptualization, data curation, formal analysis, investigation, methodology design, and preparation of the manuscript's original draft. TU provided supervision and validation, methodology design enhancement, and review and editing of the manuscript.

**Competing interests**

The authors affirm that they have no competing interests, ensuring the integrity and impartiality of the research presented in this manuscript.

**Acknowledgments**

We express our gratitude to the Japan International Cooperation Agency (JICA) for supporting the PhD scholar grant through
the Knowledge Co-Creation Program; Tiago Antonelli, Chief of the Disaster Management Division, and the Geological Survey of Brazil for providing invaluable information on the financial and administrative resources related to the risk assessment surveys. We also express our gratitude to financial support by Sabo Frontier Foundation.

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
