# Peer review of "Review article: Analysis of sediment disaster risk assessment surveys in Brazil: A critical review and recommendations"

_EGUsphere, 2024_

## Author Response (AR2)

**Respone to Referees**

**https://doi.org/10.5194/egusphere-2024-2255**

Title: Review Article: Analysis of sediment disaster risk assessment surveys in Brazil:

A critical review and recommendations

Author(s): Thiago Dutra dos Santos and Taro Uchida

We are pleased to submit the revised version of our manuscript entitled: "Review Article: Analysis of sediment disaster risk assessment surveys in Brazil: A critical review and recommendations" (DOI: <a href="https://doi.org/10.5194/egusphere-2024-2255">https://doi.org/10.5194/egusphere-2024-2255</a>). We thank the reviewers for their constructive and insightful feedback, which has significantly contributed to improving the clarity, rigor, and overall quality of the manuscript.

Due to the substantial restructuring, reorganization of paragraphs, and rewriting of multiple sections for improved clarity and corrections, the use of "track changes" resulted in an overly cluttered and unreadable document. Therefore, instead of a fully marked-up manuscript, we have prepared a point-by-point revision summary in which each significant change is documented with both the original excerpt and the corresponding revised version. This format was adopted to enhance transparency and ensure that reviewers can trace how the manuscript has been improved, without the confusion often caused by visual clutter in heavily edited documents.

We hope this approach facilitates the evaluation process. However, if the reviewers or editors prefer a version with track changes, we will promptly provide it upon request.

In response to the reviewers' comments, we have made the necessary revisions and are submitting the following materials:

- 1) Revised manuscript, in which all modifications are highlighted in blue font and accompanied by margin comments indicating the corresponding topic number, the original reviewer's remark, and our detailed response.
- 2) A point-by-point response document, organized into 27 numbered topics, each corresponding to specific issues raised by the reviewers.

We hope the revised version meets the expectations of the reviewers and the editorial board.

Please do not hesitate to contact us if any further clarification is needed.

Sincerely,

Thiago Dutra dos Santos

(on behalf of all co-authors)

**RC1: 'Comment on egusphere-2024-2255', Anonymous Referee #1, 06 Apr 2025**

**AC1: 'Reply on RC1', Thiago Santos, 11 Apr 2025**

This paper is based on an extensive review of landslide prevention measures and landslide risk assessment methods in the Federative Republic of Brazil, and is highly rated as a paper that can provide readers with broad knowledge and deep insights into landslide risk assessment methods in Brazil.

However, I would like to strongly point out that it is very difficult for readers unfamiliar with the details of these methods used in Brazil to understand the points of this paper.

1) This paper compares and discusses five methods: PMRR, GRS, GAUC, SS, and GHS. At the very least, a brief explanation of each method should be written in order to allow the reader to follow the arguments of this paper. Unless this point is clearly stated, it is difficult to properly review and comment on the results and discussion of this paper. So that I believe that a re-review is necessary.

Thank you for your valuable comments and suggestions. We agree that providing a brief explanation of each federal risk assessment method is essential for ensuring clarity and helping the reader follow the comparisons and discussions presented throughout the manuscript. To address this, we added a new section 2. This section now outlines the main objectives, scope, responsible institutions, and typical applications of the five federal surveys in Brazil. (L92 - L152 in the revised manuscript).

**Included**

**2. Overview of Federal Risk Assessment Surveys in Brazil**

Since 2004, five different federal risk assessment surveys have been conducted in Brazil, each initiated at different times. The Municipal Risk Reduction Plan (PMRR) represents Brazil's first nationwide initiative aimed at establishing a standardized framework for local-scale risk assessment and disaster mitigation planning. Developed in alignment with the National Policy for Civil Protection and Defense (PNPDEC; Law No. 12,608/2012), the PMRR promotes a paradigm shift from reactive post-disaster responses to proactive risk prevention (Mendonça et al., 2023). Its methodology involves a series of structured phases, including the assessment of geohydrological risk areas, the design of structural countermeasures, cost estimation, and structural and non-structural action plans (Alheiros, 2006). Implementation is coordinated through the Union Resources Decentralized Execution Agreement, typically executed in partnership with universities, public agencies, or private entities (e.g., UFSC, 2007; Souza et al., 2008; IPPLAN, 2016), ensuring technical rigor and local contextualization.

While the PMRR offers a comprehensive and structured framework, the Geological Risk Survey (GRS) was developed as a more responsive diagnostic tool to rapidly assess geohydrological risks in urban environments. Grounded in the conceptual understanding of risk as the interaction among hazard, vulnerability, and potential damage (Tominaga, 2012),

the GRS focuses on phenomena such as landslides, debris flows, rockfalls, floods, and flash floods (Lana et al., 2021). Supported by national legislation, it serves both as a strategic input for early warning systems at the federal level and as a technical resource for local landuse regulation, preparedness measures, and emergency response planning (Pozzobon et al., 2018). Its methodology comprises a desk-based analysis using geospatial and thematic data, followed by fieldwork to validate and classify risk areas based on terrain morphology and physical vulnerability of existing infrastructure (Pimentel et al., 2018). Although various state and municipal institutions—such as the Institute for Technological Research (IPT) in São Paulo and the Geotechnical Institute Foundation (GeoRio) in Rio de Janeiro—initially developed their own methodologies, the responsibility for standardizing and implementing the survey nationwide was later delegated to the Geological Survey of Brazil (GSB), a federal agency under the Ministry of Mines and Energy, by directive of the Civil House of the Presidency (e.g., Pascarelli et al., 2013; Lamberty & Binotto, 2022; DRM, 2023).

Whereas the GRS centers on the delineation of existing risk zones, the Susceptibility Survey (SS) seeks to anticipate where future hazards are likely to occur by evaluating the intrinsic predisposition of terrain to trigger geohydrological processes. Officially recognized in Brazil's legal framework, the SS provides municipalities with technical input to inform landuse regulation and long-term urban planning (SGB, 2023b). This methodology encompasses a range of phenomena, including landslides, debris flows, floods, and flash floods (Antonelli et al., 2020). Its primary objective is to provide municipalities with technical support for territorial management and risk mitigation strategies. The approach is grounded in geospatial modeling techniques, which integrate historical inventory data with geological, hydrological, and geomorphological variables to produce susceptibility maps and classify terrain into distinct levels (Bittar, 2014). Fieldwork is carried out for validating the modeled results. While various academic institutions have proposed alternative approaches for conducting this type of assessment, the GSB is the officially designated authority responsible for implementing SS at the national level (e.g., Lorentz et al., 2016; Dias et al., 2021).

Building upon susceptibility assessments, the Geotechnical Aptitude for Urbanization Chart (GAUC) is a technical survey designed to evaluate the suitability of terrain for supporting various forms of land use, thereby guiding safe and sustainable urban development (Antonelli et al., 2021). Intended to inform municipal planning decisions, GAUC supports territorial management, land-use regulation, and disaster risk reduction policies by identifying geotechnical favorable zones for urban expansion (SGB, 2023c). The methodology involves integrating geological, geomorphological, pedological, and topographic data with historical records, complemented by detailed field and laboratory investigations to delineate homogeneous geotechnical units suitable for urbanization (Antonelli et al., 2021). Like the previous surveys, GAUC is endorsed by Law No. 12,608/2012 and serves as a planning instrument under the National Policy for Civil Protection and Defense. Although several academic institutions contribute to GAUC development, its systematic national implementation is carried out by the GSB (e.g., Ribeiro & Dias, 2020; Polivanov et al., 2024).

Completing the Brazilian risk assessment framework, the Geological Hazard Survey (GHS) was introduced to enhance the objectivity of hazard detection and provide predictive insight into potential runout distances of sediment-related events. Developed by the GSB, the GHS uses topographic thresholds derived from statistical analyses of historical events to identify susceptible areas and estimate the potential trajectory and runout extent (Pimentel et al., 2020). It is intended to support various stakeholders—including urban planners, civil defense authorities, and policymakers—by offering standardized and spatial outputs that inform land-use regulation, emergency preparedness, and risk mitigation strategies. Its methodological structure encompasses four key stages: compilation of spatial and thematic data, identification of strategic areas, desk-based hazard modeling using topographic conditioning factors, and final field validation to assign hazard classifications (Pimentel et al., 2018). The GHS has been applied in several municipal studies to support land-use planning and disaster risk management (e.g., Facuri & De Lima Picanço, 2021; Ribeiro et al., 2021; Rocha et al., 2021).

These five federal methodologies constitute the backbone of Brazil's national strategy for assessing and managing geohydrological risks. Although they operate under a shared legal framework and pursue similar overarching goals, each survey differs substantially in scale, technical scope, practical applicability, and intended outcomes. These methodological distinctions raise critical questions regarding their complementarity, integration, and overall effectiveness in supporting disaster risk reduction (DRR) initiatives across multiple levels of governance. The following sections present a systematic comparative analysis to explore these issues, examining key dimensions such as survey design, territorial coverage, operational scale, suitability and alignment with DRR implementation, and relative cost per beneficiary.

Other points I noticed are listed below.

2) Introduction: The past knowledge or information on natural disasters in general and those on landslides disasters are mixed together. So that some organization is necessary. For example, it would be possible to first state those on natural disasters in general, and then on specific landslides disasters.

Thank you for your valuable suggestion. We agree that the initial version of the introduction mixed general disaster concepts with specific information on sediment disasters. So, in the first three paragraphs of the revised manuscript, we argued about previous studies and reports on general disaster concepts (L24 - L60 in the revised manuscript). Then, we introduce previous studies and reports on sediment disasters from the fourth to the sixth paragraphs (L61 - L91 in the revised manuscript).

**Original**

[revised manuscript text omitted]
, 2012) defines DRR responsibilities and operational phases, from risk assessment to preparedness, emergency response, recovery, and mitigation. According to Giustina (2019), the initial risk assessment phase involves identifying hazardprone areas and vulnerable populations, considering physical, environmental, and social variables. This primary stage supports subsequent steps and activities. Preparedness entails educating communities and establishing early warning systems and emergency response plans. The emergency phase focuses on immediate assistance, including evacuations, search and rescue operations, and temporary shelters. Post-disaster recovery aims to reestablish safety, rebuild infrastructure, reinforce mitigation measures, and support affected communities. The mitigation phase involves implementing strategies to reduce the likelihood and severity of such disasters through engineering solutions and land-use policies. Among the instruments supporting urban resilience in Brazil is the municipal master plan (MP), established by Brazil's Law No. 10,257 (Brasil, 2001) and modified by Law No. 12,608 (Brasil, 2012). MP is an essential document for urban planning, DRR initiatives, and longterm sustainable development strategies (MDR, 2021). MP is compulsory and required in municipalities exceeding 20,000 inhabitants.

Numerous studies have explored the aspects of Brazil's governmental DRR framework. Tominaga et al. (2012) analyzed Brazil's socioeconomic disasters and risk-management strategies. Ganem (2012), Almeida (2015), and Henrique and Batista (2020) evaluated DRR's political dimensions. Kuhn et al. (2022) discussed the evolution and impacts of DRR policies, while Alvalá et al. (2019) studied the profiles of at-risk and vulnerable populations. Silva-Rosa et al. (2015) and Matsuo et al. (2019) emphasized environmental education in disaster reduction. Mendonça and Gullo (2020) focused on societal disaster perception, Marchezini et al. (2019) on education's role in risk mitigation, and Silva and Santos (2022) on the importance of social participation in DRR. Debortoli et al. (2017) and Marengo et al. (2021) discussed the impact of climate change on Brazilian disasters, highlighting the need for climate-inclusive DRR strategies. Silva (2022) extensively analyzed federal DRR projects from 2011 to 2015, offering a broad view of government initiatives in this area. The latest Brazilian Atlas of Natural Disasters - BAND (Brasil, 2023) reports a significant rise in flood and sediment disasters from 1991 to 2023, accounting for 82% of the death toll. The

21,043 disasters have claimed 3,752 lives, left over 7.45 million homeless, and impacted nearly 77 million, either directly or indirectly. The economic cost is estimated to be approximately USD 19.63 billion.

In this study, sediment disasters refer to hazardous natural phenomena resulting from the movement, accumulation, or erosion of soil, rock, or debris materials, typically triggered by gravitational forces and/or hydrometeorological conditions (Uchida et al., 2009). Typical processes cause sediment disasters include landslides, debris flows, mudslides, rockfalls, and severe soil erosion etc. (Dai et al., 2002; Hungr et al., 2014). Sediment disasters are subject to the complex effects of two factors; natural factors, such as terrain morphology, hydrological regimes, vegetation cover, and anthropogenic activities—such as road excavations, cut-and-fill operations, unregulated urban sprawl on unstable slopes, and presence of informal settlements in high-risk zones.

Over the decades, significant global progress has been made in sediment disaster risk management, including phenomena typology and classification (e.g., Varnes, 1978; Cruden & Varnes, 1996; Hungr et al., 2001; Hungr et al., 2014; Li & Mo, 2019). Insights into the predisposing, triggering, and dynamic factors influencing these events abound (e.g., Guzzetti et al., 1999; Iverson, 2000; Dai et al., 2002; Hungr, 2007; McColl, 2022; Iverson, 2015; McDougall, 2017). Additionally, considerable advancements have been achieved in assessing these phenomena (e.g., IAEG, 1990; Corominas, 1996; Aleotti & Chowdhury, 1999; Guzzetti et al., 1999; Corominas et al., 2003; Picarelli et al., 2005; Corominas & Moya, 2008; Fell et al., 2005; 2008; Van Westen et al., 2008; Corominas et al., 2014; Davies, 2015; Hungr et al., 2016). Moreover, in recent years, innovative approaches have been developed to enhance preparedness (Colombo et al., 2005; Ayalew & Yamagishi, 2005; Uchida et al., 2011; Guzzetti et al., 2012; Chen et al., 2014; Sun et al., 2020; Di Napoli et al., 2021; Linardos et al., 2022). Despite the advancements, Maes et al. (2017) highlighted that in tropical nations, only 30% of potential risk reduction measures are recommended or implemented, with risk assessments emerging as the most frequently implemented initiatives (57%).

In Brazil, a comparative evaluation of the five federal risk assessment methodologies initiated after 2004 was conducted, including the PMRR, Geological Risk Survey (GRS), Susceptibility Survey (SS), Geotechnical Aptitude for Urbanization Charts (GAUC), and Geological Hazard Survey (GHS). Information was collected and reviewed from official guidelines and their updates (Alheiros, 2006; Brasil, 2007; Bittar, 2014; Pimentel & Dutra, 2018; Lana et al., 2021). Recently, Mendonça et al. (2023) focused exclusively on evaluating the effectiveness of the PMRR. Dias et al. (2021) conducted technical comparisons of various landslide susceptibility mapping methods, including the official SS, and several academic approaches. Rocha et al. (2021) argued the effectiveness of SS, GAUC, and GHS based on the case studies in Nova Friburgo, Rio de Janeiro state. However, no previous studies have undertaken a systematic and comparative analysis encompassing all five federal risk assessment methodologies currently implemented in Brazil. Moreover, the existing literature has not thoroughly examined the methodological components, national coverage, their suitability to inform and support DRR initiatives, and the cost per beneficiary. This study

aims to bridge these critical gaps by offering a comprehensive evaluation of each federal survey, identifying methodological deficiencies, and proposing evidence-based improvements to enhance the Brazilian DRR strategies for a more resilient society.

**3) Line 31: What is "sediment disaster"? Write the definition.**

According to the reviewer's comments, we added the definition of sediment disasters. As follow:

**Included**

L61 - L67 in the revised manuscript — "In this study, sediment disasters refer to hazardous natural phenomena resulting from the movement, accumulation, or erosion of soil, rock, or debris materials, typically triggered by gravitational forces and/or hydrometeorological conditions (Uchida et al., 2009). Typical processes cause sediment disasters include landslides, debris flows, mudslides, rockfalls, and severe soil erosion etc. (Dai et al., 2002; Hungr et al., 2014). Sediment disasters are subject to the complex effects of two factors; natural factors, such as terrain morphology, hydrological regimes, vegetation cover, and anthropogenic activities—such as road excavations, cut-and-fill operations, unregulated urban sprawl on unstable slopes, and presence of informal settlements in high-risk zones."

**4) Line 76: What is "the core elements"? What kind of elements did the previous studies deal with?**

**Original**

However, no previous studies have exclusively focused on the core elements of federal risk assessment surveys nationwide. Hence, this study aims to bridge this gap by analyzing the nuances of these surveys, identifying methodological deficiencies, and proposing improvements, thus enhancing Brazilian strategies for a more resilient society.

**Revised**

We revised the text to clearly express which core elements are lacking in the current literature. The final paragraph was rewritten as follows:

L79 - L91 in the revised manuscript — "In Brazil, a comparative evaluation of the five federal risk assessment methodologies initiated after 2004 was conducted, including the PMRR, Geological Risk Survey (GRS), Susceptibility Survey (SS), Geotechnical Aptitude for Urbanization Charts (GAUC), and Geological Hazard Survey (GHS). Information was collected and reviewed from official guidelines and their updates (Alheiros, 2006; Brasil, 2007; Bittar, 2014; Pimentel & Dutra, 2018; Lana et al., 2021). Recently, Mendonça et al. (2023) focused exclusively on evaluating the effectiveness of the PMRR. Dias et al. (2021) conducted technical comparisons of various landslide susceptibility mapping methods, including the official SS, and several academic approaches. Rocha et al. (2021) argued the effectiveness of SS, GAUC, and GHS based on the case studies in Nova Friburgo, Rio de Janeiro state. However, no previous studies have undertaken a systematic and comparative analysis encompassing all five federal risk assessment methodologies currently implemented

in Brazil. Moreover, the existing literature has not thoroughly examined the methodological components, national coverage, their suitability to inform and support DRR initiatives, and the cost per beneficiary. This study aims to bridge these critical gaps by offering a comprehensive evaluation of each federal survey, identifying methodological deficiencies, and proposing evidence-based improvements to enhance the Brazilian DRR strategies for a more resilient society."

**5) Line 120: ...disaster reduction... --> ...disaster risk reduction...?**

**Original**

**2.3.2 Applicability for disaster reduction countermeasures**

**Revised**

L199 in the revised manuscript – Update the manuscript to enhance clarity and standardization. Disaster Risk Reduction was adopted.

- 2.3.2 Applicability for disaster risk reduction countermeasures
- 6) Line 125: There are two "Local Scale"s.

**Original**

We adopted the ordinal classification system, proposed by Fell et al. (2008) and Corominas et al. (2014), for scoring, emphasizing customization of assessment scale and scope to match local and state authorities' requirements, data availability, and survey objectives. Consequently, we categorized the surveys into four distinct assessment scales: National Scale (< 1: 250,000), Regional Scale (1: 250,000 to 1: 5,000), Local Scale (1: 25,000 to 1: 5,000), and Local Scale (1: 25,000 to 1: 5,000).

**Revised**

L204 in the revised manuscript – Removed the repeated local scale. And included the corrected Plot scale (< 1: 5,000).

We adopted the ordinal classification system proposed by Fell et al. (2008) and Corominas et al. (2014), emphasizing the need to tailor assessment scales and scopes according to institutional needs, data availability, and the specific objectives of each survey. Based on this framework, we grouped the surveys into four categories: National Scale (< 1: 250,000), Regional Scale (1: 250,000 to 1: 25,000), Local Scale (1: 25,000 to 1: 1,000), and Plot scale (< 1: 5,000).

**7) Line 126-127: This sentence seems to be very difficult to understand. I would like it to be written more clearly.**

**Original**

These scales enable a tailored approach, allowing specific interventions at each level and ensuring practical and effective outcomes for diverse contexts.

**Revised**

Thank you for your observation. We agree that the original sentence required greater clarity and have reformulated the passage to enhance its readability and explicability. The revised section now clearly outlines the categorization of the surveys and the rationale behind the assessment scales. However, we are not entirely sure if we have fully captured the intent of your comment. If the updated version still does not adequately address your concern, we would be grateful if you could provide further clarification so we can make the necessary improvements.

L201–L207 in the revised manuscript – "We adopted the ordinal classification system proposed by Fell et al. (2008) and Corominas et al. (2014), emphasizing the need to tailor assessment scales and scopes according to institutional needs, data availability, and the specific objectives of each survey. Based on this framework, we grouped the surveys into four categories: National Scale (< 1: 250,000), Regional Scale (1: 250,000 to 1: 25,000), Local Scale (1: 25,000 to 1: 5,000), and Plot scale (< 1: 5,000). Assuming that each disaster prevention initiative requires a suitable assessment scale, we analyzed how well each survey aligns with these initiatives by applying an applicability gradient—from 0 (Not applicable, dark blue) to 4 (Fully applicable, dark red)—to systematically evaluate their relevance."

8) Line 143-144: Is "structural measure (SC)" included in the "six non-structural initiatives"? It seems obviously contradictory.

**Original**

These comprise six nonstructural initiatives promoting appropriate land-use policies (MP, LULOL, and SL), management (EWS, ERP), and one structural measure (SC).

**Revised**

L222–L223 in the revised manuscript – "These comprise five nonstructural initiatives promoting appropriate land-use policies (MP, LULOL, and SL), management (EWS, ERP), and one structural measure (SC)."

9) Line 177: "cost-benefit ratio" --> "cost per beneficiary"?

**Original**

A higher cost–benefit ratio....

**Revised**

Systematically adjusted the terminology related to cost per beneficiary to ensure consistency and standardization.

L256 in the revised manuscript – "A higher cost per beneficiary..."

10) Figure 1.: At the top are figures comparing the number of disasters per state with total area, urban area, population and so on. I would like a clear explanation of how the number of disasters is counted. If 10 landslides occur in one heavy rainfall

**event, should each be counted as one, or should they be counted as ten? This would likely change the interpretation of the figures.**

It is essential to clarify that the disaster counts in this database (Brasil, 2023) are based on officially recognized disaster declarations, not on individual occurrences of phenomena. In other words, a single event recorded in the Atlas may encompass multiple landslides triggered by the same meteorological episode. Therefore, the number of disasters reflects the number of declared emergencies rather than the total number of individual landslide occurrences.

**Included**

L162–L165 in the revised manuscript – Enhanced version included in the methods section (2.1 Data collection and analysis). As follows:

"In this database, disasters are recorded based on the issuance of official emergency or disaster declarations, rather than on the count of individual physical phenomena. For example, a single entry may represent one or several landslides that occurred during the same rainfall event. Therefore, the disaster count in this study reflects the number of formally recognized events at the municipal level, not the total number of landslide occurrences."

**11) What is "critical municipalities"? It means the 286 municipalities? Now that the explanation appears in the latter part, you should add some explanation on it before readers see this figure.**

We have clarified the definition of "critical municipalities" in *Section 2.1 (Data Collection and Analysis)*, where we explain that this designation is based on federal risk classifications and specify that the total number of municipalities included in our study is 821, based on the most recent available data.

**Included**

L167–L170 in the revised manuscript – "The municipalities highly susceptible to sediment-related disasters were retrieved from the Ministry of Regional Development—National Secretariat of Civil Protection and Defense (MDR, 2012), which designates these locations as "critical municipalities" due to their elevated risk levels. Initially, 286 cities were identified under this classification, the number was later expanded to 821 based on updated federal reports. In this study, we utilized the most recent data, comprising 821 critical municipalities, for our analysis. These areas have been prioritized for the implementation of DRR assessment surveys."

**12) Figure 2: Why is this figure just for PMRR, GRS, and SS? Why are not GHS and GAUC shown?**

Thank you for bringing this important point to our attention. The decision to include only PMRR, GRS, and SS in Figure 2 was based on the extent of their municipal coverage across Brazil. While GAUC and GHS are indeed part of our assessment framework, they were excluded from this specific figure due to their limited representativeness. According to Table

1, GAUC and GHS assessments have been conducted in only 17 and 12 municipalities, respectively—a notably small sample size when compared to the broader implementation of the other methods.

**Included**

320–L321 in the revised manuscript – So, we add this in the text as follows:

"GAUC and GHS were excluded from this figure due to the small number of municipalities implemented (17 and 12, respectively)."

13) Line 195: "rho" should be written in Greek letter.

**Original**

Given the nonlinear nature of these relationships, we employed the Spearman coefficient (rho) to conduct a nonparametric data analysis.

L274 in the revised manuscript – The term "rho" has been updated to the corresponding Greek symbol ( $\rho$ ) in the revised manuscript to ensure proper formatting and consistency with academic standards.

**Revised**

"Given the nonlinear nature of these relationships, we employed the Spearman coefficient  $(\rho)$  to conduct a nonparametric data analysis."

**14) Line 292-293: "plot scale" and "partial plot" How large are they?**

We agree that the terms "plot scale" and "partial plot" require further clarification. We added the size of each scale as follows:

**Included**

L351–L353 in the revised manuscript – "The topographic units used to assess risk vary depending on the purpose of each survey. The SS, GAUC, and GHS conduct catchment (> 10 ha) analyses (Table 1; Fig. 3). In some instances, GHS also conducts plot scale  $(1 - 100 \text{ m}^2)$  analyses. On the other hand, the PMRR and GRS employ partial plot  $(100 - 500 \text{ m}^2)$  and hillslope (>  $500 \text{ m}^2 - 10 \text{ ha}$ ) examinations."

15) Line 345-347: If it is written in the literature, the accuracy of the prediction should be evaluated not only in terms of the hit rate but also in terms of the miss rate. The GHS method may have determined in advance that 95% of the collapsed areas were dangerous, but it would be appropriate to also indicate how many slopes were determined to be dangerous but did not collapse.

**Original**

The study found that the GHS method outperformed the others, achieving an accuracy coefficient of 95% in identifying areas destroyed by the disaster.

According to the reviewer's comment, we revised the original article about the missed rate, but it was not directly shown in the previous study. On the other hand, we did find out what range was considered to be the dangerous range, so we have added that information as follows.

**Revised**

L403–L406 in the revised manuscript – "However, the authors also reported that 47% of areas were classified as hazardous by the GHS, suggesting that there were also many slopes that were deemed dangerous but did not collapse."

**16) Line 376: Where is Figure 7?**

**Original**

Figure 7 illustrates the distribution of municipalities across the Brazilian states....

L435 in the revised manuscript – Thank you for catching this error. The reference to Figure 7 was an error introduced during the final editing stage and should be corrected to refer to Figure 4.

**Revised**

"Figure 4 illustrates the distribution of municipalities across the Brazilian states...."

17) Line 384-402: The percentage values in the text cannot be found in Figure 4. Please either add a figure or discuss only the values that can be found in the figure.

**Original**

The Federal District and states like Rio de Janeiro (42%), Acre (40%), Espírito Santo (39%), and Pernambuco (37%) lead in integrating such strategies. Conversely, states including Tocantins, Rondônia, Amazonas, Amapá, and others show less than 20% incorporation of these preventive measures in their MPs, highlighting regional disparities in DRR emphasis (Fig. 4).

The northern region shows particular deficiencies in disaster prevention, with states like Roraima and Rondônia lagging significantly; 80% and 77% of their municipalities, respectively, lack additional disaster prevention strategies. In the Northeast, disparities are stark, with low implementation rates in states like Piauí, Rio Grande do Norte, and Paraíba. Pernambuco stands out in terms of more comprehensive DRR efforts. In contrast, the Southeastern region displays the highest DRR engagement in Brazil, with Minas Gerais mapping 30% of risk areas but only 5% implementing protective measures. Espírito Santo and Rio de Janeiro demonstrate higher implementation rates in various DRR strategies, whereas São Paulo faces challenges, with 54% of its municipalities not implementing any measure (Fig. 4). Santa Catarina leads the Southern region in disaster risk mapping, yet lags in engineering and EWS implementations. Paraná shows moderate DRR implementation,

and Rio Grande do Sul has the lowest engagement. The Distrito Federal excels in DRR strategies in the Middle-Western region, although it lacks engineering plans. The DRR efforts in Mato Grosso and Goiás are inadequate, and Mato Grosso do Sul shows a mixed performance, emphasizing the regional challenges in DRR strategy implementation (Fig. 4).

Thank you for pointing out this incredible mistake. We have revised the text to ensure that all percentage values now directly correspond to those presented in Figure 4. Additionally, the graphs have been improved to visually emphasize regional divisions through the use of color coding, enhancing interpretability and regional comparison in *Section 5.2: Examining the correlation between risk assessment surveys and the implementation of disaster reduction countermeasures*.

**Revised**

L433-L475 in the revised manuscript – Risk assessment surveys are vital resources for various risk-management initiatives. Therefore, the effectiveness of these surveys can be evaluated by examining the activities and initiatives developed from the basic information provided by them. Figure 4 illustrates the distribution of municipalities across the Brazilian states that have adopted various DRR initiatives, such as master plans (MP), landslidespecific laws (SL), land-use and land-occupation laws (LULOL), early warning systems (EWS), emergency response plans (ERP), and structural countermeasures (SC). The regional distribution of DRR initiatives across Brazilian states reveals notable contrasts in implementation levels. First, excluding the Federal District, the implementation of landslidespecific laws (LSL) remains notably low across all states. Only Rio de Janeiro and Pará exceed the 5% threshold, standing out as the exceptions in this category. In the Northern region, most states exhibit relatively low adoption of disaster risk reduction measures. However, Amazonas, Pará, and Amapá present higher percentages in certain indicators in this region. Amazonas, for instance, shows considerable efforts in implementing emergency response plans (23%) and early warning systems (11%). Pará demonstrates moderate values across all initiatives, particularly in master plans (13%) and LULOL (12%). Amapá also stands out with 31% of municipalities having ERP and 19% implementing LULOL. In contrast, Roraima and Tocantins register the lowest levels in the region, with most indicators below 5%, and complete absence of early warning systems, emergency plans, and structural countermeasures in Roraima. In the Northeastern region, the implementation pattern is more heterogeneous. States such as Pernambuco and Alagoas lead in most indicators. Pernambuco exhibits significant adoption of master plans (20%), emergency response plans (25%), and structural countermeasures (11%), while Alagoas shows high percentages in early warning systems (14%) and ERP (18%). Other states, such as Ceará and Bahia, demonstrate moderate values across all initiatives. In contrast, Piauí and Paraíba appear among the least engaged in the region, with consistently low percentages for specific plans, early warning systems, and structural countermeasures. In the Midwestern region, results vary significantly. The Federal District represents a clear outlier, reporting 100% implementation for all DRR categories except for structural countermeasures. Mato Grosso do Sul follows with comparatively high adoption of master plans (13%), LULOL (14%), and EWS (8%). Meanwhile, Mato Grosso and Goiás exhibit limited implementation, with most indicators—particularly EWS, ERP, and SC—remaining below 5%.

Figure 4: Percentage of municipalities implementing sediment-related DRR initiatives in each Brazilian state. Values represent the proportion of municipalities (%) per state. Data source: IBGE (2020) DRR dataset. Bars are color-coded by Brazil's five macro-regions."

The Southeastern region stands out as the most advanced in DRR implementation. Rio de Janeiro and Espírito Santo lead the country, with exceptionally high percentages across nearly all indicators. Rio de Janeiro, for example, reports 77% of municipalities with ERP, 41% with EWS, and 30% with SC. Espírito Santo shows similar results, including 63% ERP and 26% EWS. São Paulo and Minas Gerais also demonstrate widespread adoption, with São Paulo exceeding 10% in all indicators and Minas Gerais registering 20% for ERP and 18% for SC. In the Southern region, DRR measures are generally well adopted. Paraná shows the highest percentages for master plans (31%) and LULOL (31%) among all states in the region. Santa Catarina also performs well, particularly in EWS (16%) and ERP (32%). Rio Grande do Sul, while displaying lower values compared to its southern counterparts, still achieves notable implementation for ERP (30%). Overall, the Southeast and South regions exhibit the highest concentration of municipalities with DRR measures, while the North and Midwestern—excluding the Federal District—tend to lag behind, with considerable disparities within and between regions."

18) Line 398: "Santa Catarina leads to... EWS implementations." What is this sentence based on? I cannot find any evidences in Figure 4 or others.

**Original**

Santa Catarina leads the Southern region in disaster risk mapping, yet lags in engineering and EWS implementations.

Thank you for pointing this out. The sentence referring to Santa Catarina as a leader in EWS implementation has been removed from the revised manuscript. This adjustment was made to ensure consistency with the data presented in Figure 4. As explained in our response to Comment 17, the entire paragraph was updated to accurately reflect only the values available in Figure 4.

**Revised**

L471–L475 in the revised manuscript – "Santa Catarina also performs well, particularly in EWS (16%) and ERP (32%). Rio Grande do Sul, while displaying lower values compared to its southern counterparts, still achieves notable implementation for ERP (30%). Overall, the Southeast and South regions exhibit the highest concentration of municipalities with DRR measures, while the North and Midwestern—excluding the Federal District—tend to lag behind, with considerable disparities within and between regions."

**19) Figure 6: The correspondence with the six initiatives written in section 2.3.3 is unclear.**

**Original**

A relevance matrix (Fig. 6) can help to understand the applicable range of these surveys (Hungr et al., 2005; Fell et al., 2008; Corominas et al., 2014). The color gradient, ranging from dark blue (0—Not applicable) to dark red (4—Fully applicable), indicates the level of applicability of each element according to the survey scales.

The PMMR and GRS can be applied to various initiatives, existing urban developments, enhancing EWSs, formulating emergency plans, and disseminating practical information (Fig. 6). However, their application is less prevalent in urban planning. Furthermore, while they may be used to support local legislative frameworks, they are not recommended for supporting national legislative policies.

The SS and GAUC can apply for similar initiatives, such as information communication and dissemination, urban planning and development, enhancing warning systems, and implementing climate change countermeasures, but they are not commonly employed in developing emergency plans. Although the SS is not applicable for prioritizing SC, it is highly relevant to the legislative framework at both the national and municipal levels. In contrast, the GAUC is not commonly used for national legislative policies. GHS achieves the widest applicability. This survey can apply all initiatives shown in Fig. 6, although some are partially applicable, especially in framing national policies.

Based on the reviewer's comments, we have reconsidered Figure 6. Disaster prevention initiatives are extremely diverse, making it challenging to encompass them all in a single figure. In this study, we believe that following the reviewer's comments and organizing the

six initiatives used in the previous section will improve the consistency of this manuscript and facilitate readers' understanding. We revised the figure and the main text accordingly.

**Revised**

L496–L516 in the revised manuscript – "To complement the operational analysis presented in Section 5.2, a relevance matrix (Fig. 6) was developed to explore the applicability of federal risk assessment surveys across multiple dimensions of disaster risk reduction. While the previous section focused on the implementation status of key DRR initiatives based on official indicators from the municipal profiles (IBGE, 2020), the matrix presented here evaluates the suitability of these DRR derived from internationally recognized methodological frameworks (Hungr et al., 2005; Fell et al., 2008; Corominas et al., 2014).

Figure 6: Brazilian DRR assessment relevance matrix based on applicability recommendations from Hungr et al. (2005), Fell et al. (2008), and Corominas et al. (2014).

The matrix displays the degree to which each survey supports different DRR elements, using a gradient scale from dark blue (0—Not applicable) to dark red (4—Fully applicable). This classification reflects the functional alignment of each survey with best practices for its respective scale, taking into account its defined scope and the extent to which it is integrated into formal governance practices. The resulting overview highlights distinct differences in applicability among the methodologies. The matrix reveals a clear differentiation in the breadth and depth of applicability among the five federal risk assessment methodologies. The PMRR and GRS exhibit consistently high applicability across a range of DRR initiatives, particularly in emergency response planning (ERP), early warning systems (EWS), and structural countermeasures (SC). Their operational versatility enables integration into a broad set of initiatives; however, their role in shaping legislative frameworks—particularly LSL and LULOL—remains limited. In contrast, the GAUC demonstrates strong alignment with legal instruments, though its contribution to ERP appears comparatively constrained. The SS similarly supports the legislative dimension, but its applicability is markedly lower in ERP and SC. Finally, the GHS stands out as the most applicable methodology, achieving either full (score 4) or substantial (score 3) relevance across all DRR categories. Its balanced integration emphasizes its utility as a comprehensive tool for multi-level risk governance."

**20) Line 443: cost-benefit ratio --> cost per beneficiary?**

**Original**

The GRS, which is faster and more cost-effective than PMRR, achieved a cost-benefit ratio of 1.309, which showcased its economic and practical approach.

We corrected it. Cost per beneficiary.

**Revised**

L524–L526 in the revised manuscript – "Notably, GRS is faster and less resource-intensive than the PMRR method, achieving a cost per beneficiary of 1.309. This result highlights the practical applicability and economic advantages of this approach in disaster risk reduction efforts."

**21) Line 447: \$0.0004 per beneficiary ... \$0.009 in Table.5 - Which is correct?**

**Original**

...\$0.0004 per beneficiary, demonstrating highly effective resource use.

Thank you for pointing this out. The correct value is \$0.009 per beneficiary, as indicated in Table 5. The discrepancy in the text has been corrected accordingly.

**Revised**

L530 in the revised manuscript – "...\$0.0009 per beneficiary, demonstrating highly effective resource use."

**RC2: 'Reply on AC1', Anonymous Referee #1, 29 Apr 2025**

Thank you for your sincere response to my comments. However, it hasn't been uploaded the revised manuscript, yet. I would like to see the revised version. Could you please let me see it?

**AC2: 'Reply on RC2', Thiago Santos, 01 May 2025**

Dear Referee,

Thank you again for your thoughtful comments and interest in reviewing our revised manuscript.

I sincerely appreciate your request to see the updated manuscript. To provide you with an appropriate response, I consulted the editorial office regarding the possibility of uploading a revised version at this stage. However, they clarified that, since the manuscript is still under open discussion, the journal's policy does not allow authors to upload revised versions until the end of the discussion period. As indicated in the peer-review process description, authors are expected to respond to all comments first, and only then may they be invited by the editor to submit a formal revision.

To address your request as much as possible within the current format, I have revised and included below the updated version of the specific section you previously commented on, particularly points 1, 17, and 19. I hope this partial revision clarifies how your suggestions are incorporated into the manuscript.

Additionally, I have uploaded a supplementary file in this discussion thread with a more suitable format (PDF) to make it easier for you to review the changes in context.

Thank you once again for your constructive feedback and understanding. I remain fully committed to incorporating all your suggestions in the final revised manuscript as soon as the editor officially requests it.

Please don't hesitate to let me know if you have any further requests or suggestions.

Best regards,

Thiago Santos

Supplementary information was provided.

RC3: 'Reply on AC2', Anonymous Referee #1, 02 May 2025

AC3: 'Reply on RC3', Thiago Santos, 03 May 2025

I was not aware that the journal's policy does not allow authors to upload revised versions until the end of the discussion period. I apologize and thank you very much for your kind assistance.

Thanks to you, I was able to read a brief explanation of PMRR, GRS, GAUC, SS, and GHS, and was able to easily read through the Discussion chapter.

In addition, I have a few small comments as shown below.

**22) Figure 4 caption: "Distribution of municipalities in Brazilian state..." --> "The number of municipalities in Brazilian state..."?**

**Original**

Figure 4: Distribution of municipalities in Brazilian states implementing sediment-related DRR initiatives (in percentages). Analysis based on IBGE (2020) DRR dataset.

We revised the figure caption to improve clarity regarding the meaning of the values presented. This ensures that readers clearly understand that the analysis reflects relative (not absolute) coverage of DRR initiatives across states.

**Revised**

L463–L465 in the revised manuscript – "Figure 4: Percentage of municipalities implementing sediment-related DRR initiatives in each Brazilian state. Values represent the proportion of municipalities (%) per state. Data source: IBGE (2020) DRR dataset. Bars are color-coded by Brazil's five macro-regions."

23) Figure 5: What exactly do the vertical and horizontal axes in Figure 4 mean? Please add labels for them. For example, I imagined that the graph at the top left had the vertical axis representing the percentage of cities in the state that have MMP, and the horizontal axis representing the number of cities in the state that have implemented PMRR. Is this correct? If yes, whether this understanding is correct or not, I think it would be better if you added an explanation to the main text so that readers can understand accurately.

**Original**

Figure 5: Spearman correlation displaying the relationship between the most important DRR initiatives and risk assessment surveys across 5570 municipalities. Analysis based on IBGE (2020) and SGB (2023a, b).

Thank you for the valuable comment regarding Figure 5. You are correct: in each panel, the horizontal axis shows the number of municipalities per state implementing each risk assessment survey (PMRR, GRS, SS), and the vertical axis shows the number of municipalities per state adopting each specific DRR strategy (*e.g.*, Municipal Master Plan, Local LULC, Local SL, EWS, ERP, SC). Both axes represent absolute counts of municipalities, not percentages.

**Revised**

To improve clarity, we did:

1) Add a sentence in the results section to guide the reader:

L478–L480 in the revised manuscript – "As shown in Figure 5, the analysis considers absolute counts of municipalities per state for risk assessment surveys and DRR initiatives,

providing a robust measure of their association. Because the GAUC and GHS have low implementation levels, they were removed from this analysis."

2) Modify the figure caption to explain that both axes show the number of municipalities per state.

L482–L484 in the revised manuscript – "Figure 5: Spearman's rank correlation between the number of municipalities per state implementing risk assessment surveys (PMRR, GRS, SS) and the number of municipalities per state adopting specific DRR strategies (Municipal Master Plan, Local LULOL, Local SL, EWS, ERP, SC). Analysis includes 5570 Brazilian municipalities based on IBGE (2020) and SGB (2023a, b)."

3) Add explicit axis labels to the figure.

Spearman's coefficient (ρ). Asterisks denote significant correlations: \* p value <.05, \*\* p value <.01, \*\*\* p value <.01. The gray area represents the 95% confidence interval

**24) L515: Just for the GHS survey? Other surveys could be improved if frequency and magnitude analysis are incorporated or enhanced.**

**Original**

Incorporating frequency and enhancing magnitude analyses in the GHS survey is essential for improving hazard assessment and prediction (Table 2).

Thank you for this valuable comment. We agree that incorporating frequency and magnitude analyses would also benefit the other surveys. We have revised the text to clarify that these improvements are not limited to the GHS survey, but could strengthen hazard assessment across multiple surveys. We appreciate this insight, which helped improve the generalizability of our recommendations.

**Revised**

L597– L601 in the revised manuscript – "Incorporating frequency analyses and enhancing magnitude assessments are crucial for improving the GHS survey and advancing the overall effectiveness of quantitative risk assessment and prediction across the other surveys (Table 2)."

25) L553: 14 %? Does this mean the average percent of all the states shown in the Figure 4? If yes, some explanation is necessary in the text or in the figure.

**Original**

The results showed low adherence to surveys in the MPs (14%) (Fig. 4).

We confirm that the 14% refers to the overall percentage of Brazil's 5,570 municipalities that have implemented Municipal Master Plans (MPs), as shown in Figure 4. This aggregate information is not explicitly labeled in the figure itself. Therefore, we included this clarification directly in the results section to ensure that readers understand the origin and meaning of this value, as you suggest.

**Included**

Results section: L438–L444 in the revised manuscript – "First, the Federal District was excluded because it contains only one municipality, which could distort the overall analysis. The results showed that among Brazil's 5,570 municipalities, only about 15% on average have implemented master plans. This low implementation rate is consistent across most states. The Southeast and South regions demonstrate higher implementation levels (Fig. 4), while the North and Midwestern regions show considerably lower levels. Rio de Janeiro (33%), Espírito Santo (27%), and Santa Catarina (23%) lead in MP implementation, whereas states such as Tocantins (4%), Rondônia, Amazonas, Amapá, Piauí, and Paraíba (6%) fall below the national average."

**Revised**

Discussion section: L635–L637 in the revised manuscript – "The results highlighted low adherence to master plan implementation, with approximately 15% of Brazil's 5,570 municipalities reporting implementation (Fig. 4)."

26) L553-554: Is the strong correlation seen in Figure 5 just seen in the disaster-affected municipalities? In other words, did the parent-set of data shown in Figure 5 include only disaster-affected municipalities?

**Original**

The strong correlation between legislative prevention mechanisms (MP, LULOL, SL) and surveys (Fig. 5) indicates a trend toward implementing legal measures in disaster-affected municipalities.

Thank you for this important question. We clarify that the data presented in Figure 5 include all 5,570 Brazilian municipalities, not only those affected by disasters. Therefore, the strong correlations observed between legislative prevention mechanisms (MP, LULOL, SL) and surveys are based on the whole national dataset, regardless of whether municipalities have experienced past disaster events.

We recognize that the original discussion sentence could imply that the analysis was restricted to disaster-affected municipalities. To improve clarity, we revised the discussion sentence.

**Revised**

L637–L639 in the revised manuscript – "The strong correlation between legislative prevention mechanisms (MP, LULOL, SL) and surveys (Fig. 5) indicates a general trend toward implementing legal measures across municipalities, which higher implementation rates may partly influence in disaster-affected areas."

27) L603-604: As far as seeing the new Fig.6, PMRR and GRS are both better than GAUC. I hope my comments are helpful to you.

**Original**

In contrast, the GAUC, introduced in 2014, and the GHS, launched in 2018, demonstrate high applicability in DRR initiatives.

Thank you for this important observation. We agree that, as shown in the updated Figure 6, PMRR and GRS rank higher in applicability than GAUC. We have revised the discussion section to clarify this point and to reflect the relative performance of all surveys better. Additionally, we recognize that GAUC has a distinct purpose compared to the other surveys analyzed, as it focuses primarily on urban planning in non-consolidated safety areas rather than exclusively on disaster prevention. This distinct scope may partially explain its more moderate performance in the DRR applicability ranking. We appreciate this comment, which allowed us to improve the precision and nuance of our interpretation.

**Revised**

L686–L692 in the revised manuscript – Despite these strengths, the applicability of SS is considered moderate to low compared to other surveys (Fig. 6). Notably, PMRR and GRS rank high in applicability. In contrast, while demonstrating moderate potential, the GAUC, introduced in 2014, serves a distinct purpose focused on urban planning in non-consolidated safety areas, which may limit its performance in DRR-specific strategies. Finally, the GHS, launched in 2018, exhibits the highest applicability in DRR initiatives. However, GAUC and GHS present challenges, including moderate execution times, inherent costs, and require more detailed analyses (Antonelli et al., 2021; Pimentel & Dutra, 2018). Due to these complexities, they incur moderate to high costs per beneficiary and show relatively low adherence among practitioners.

**RC4: 'Reply on AC2', Anonymous Referee #1, 02 May 2025**

Finally, I believe that no further peer review is necessary.

**AC4: 'Reply on RC4', Thiago Santos, 03 May 2025**

Thank you for your message. We truly appreciate your constructive feedback and the time you dedicated to reviewing our manuscript.

**RC5: 'Comment on egusphere-2024-2255', Anonymous Referee #2, 08 May 2025**

Dear Authors,

I read your manuscript about various landslide risk assessment surveys conducted at the federal level in Brazil. The manuscript analyzes five main survey approaches (the Municipal Risk Reduction Plan (PMRR), the Geological Risk Survey (GRS), the Susceptibility Survey (SS), the Geotechnical Aptitude for Urbanization Charts (GAUC), and the Geological Hazard Survey (GHS)), for each of which shortcomings and range of applicability have been analyzed. I also read the comments of previous reviewer and your answers, along with the modifications you made based on their suggestions. You already addressed all the suggestions I could have made, hence, I believe that your manuscript can be processed for publication without further revisions.

**AC5: 'Reply on RC5', Thiago Santos, 10 May 2025**

Dear Reviewer #2,

Thank you very much for your evaluation and for dedicating your time to review our manuscript.

We sincerely appreciate your kind comments and your recognition of the revisions we made in response to Referee #1's suggestions. These changes have strengthened the manuscript, making it clearer and more robust.

We are grateful for your support and encouraged by your positive feedback.

Finally, we hope that this article encourages a critical and constructive dialogue within Brazil's scientific and technical community, aiming to strengthen methodological approaches to quantitative risk assessment. By reflecting on existing federal frameworks, our intention is to support their refinement and enhance their effectiveness in disaster risk prevention strategies.

Best regards,

On behalf of all co-authors

Thiago Dutra dos Santos